# AIReg-Bench: Benchmarking Language Models That Assess AI Regulation Compliance

## Abstract

As governments move to regulate AI, there is growing interest in using Large Language Models (LLMs) to assess whether or not an AI system complies with a given AI Regulation (AIR). However, there is presently no way to benchmark the performance of LLMs at this task. To fill this void, we introduce **AIReg-Bench**: the first benchmark dataset designed to test how well LLMs can assess compliance with the EU AI Act (AIA). We created this dataset through a two-step process: (1) by prompting an LLM with carefully structured instructions, we generated 120 technical documentation excerpts (samples), each depicting a fictional, albeit plausible, AI system — of the kind an AI provider might produce to demonstrate their compliance with AIR; (2) legal experts then reviewed and annotated each sample to indicate whether, and in what way, the AI system described therein violates specific Articles of the AIA. The resulting dataset, together with our evaluation of whether frontier LLMs can reproduce the experts' compliance labels, provides a starting point to understand the opportunities and limitations of LLM-based AIR compliance assessment tools and establishes a benchmark against which subsequent LLMs can be compared. The dataset and evaluation code are available at https://anonymous.4open.science/r/aireg-bench-5259/ .

## 1 Introduction and Problem Statement

Across the world, AI Regulation (AIR) initiatives are either under development or have graduated the legislative process and gone into effect (Sloane & Wüllhorst, 2025; Chun et al., 2024; Alanoca et al., 2025). For both the regulators who enforce these regulations and the regulated parties who must comply with them, *compliance assessments*, whereby an AI system is evaluated for its compliance with respect to an AIR, play a pivotal role (Mökander et al., 2021; Ada Lovelace Institute, 2024; Anderljung et al., 2023; Raji et al., 2022; Reuel et al., 2024a). For example, the European Union's AI Act (AIA) — dubbed "the world's first comprehensive AI law" (European Parliament, 2024) — requires that providers of high-risk AI systems conduct such an assessment before putting their products on the market in the EU (EU, 2024, Art. 43).

Despite their importance, however, certain AIR compliance assessments remain costly and time-consuming (Koh et al., 2024; Costanza-Chock et al., 2022; Sovrano et al., 2025). For example, some estimate that these assessments can take up to two-and-a-half days (European Commission, 2021) and cost EUR 7,500 for each AI system (Haataja & Bryson, 2021), accounting for up to 17% of the total expense of an AI project (Laurer et al., 2021). These high costs may contribute to a level of regulatory overhead that some have called unsustainable for AI providers and regulators alike (Laurer et al., 2021; Gikay, 2024; Reuel et al., 2024b; Koh et al., 2024; Molnar, 2024; Micklitz & Sartor, 2025) and, since it disproportionately affects small and medium-sized enterprises due to their lower resources (Stampernas & Lambrinoudakis, 2025), a potential hazard for fair competition (Martens, 2024; Gazendam et al., 2023; Wu & Liu, 2023; Guha et al.; Iliasova et al., 2025).

This may help explain why there is growing interest in, and experimentation with, using Large Language Models (LLMs) to perform (or, at least, streamline) AIR compliance assessments (Micklitz & Sartor, 2025; Li et al., 2025; Sovrano et al., 2025; Davvetas et al., 2025; Kővári et al., 2025; Makovec et al., 2024; Marino et al., 2024). And yet, there is still no standardized method for quantitatively evaluating and comparing the performance of LLMs at this particular task, creating uncertainty about the extent to which LLMs can be entrusted with it (Davvetas et al., 2025).

To fill this void, we present AIReg-Bench: an open dataset for benchmarking the performance of LLMs at AIA compliance assessments. This dataset, which is available at https://anonymous.4open.science/r/aireg-bench-5259/ , consists of 120 technical documentation excerpts (i.e., details on system development and testing procedures) (Sovrano et al., 2025; Königstorfer & Thalmann, 2022). Each one provides information about a fictional, albeit plausible, AI system — specifically, a high-risk AI (HRAI) system under the AIA's risk-based approach (EU, 2024, Art. 6). The samples in this dataset (i.e., the excerpts) are generated by an LLM-based technique, described in Section 2, allowing us to create diverse samples efficiently and at scale. As outlined in Section 3, each sample is then labeled by legal experts to indicate whether, and in what way, the system described therein violates specific Articles of the AIA.

To showcase AIReg-Bench at work, we evaluated 10 frontier LLMs. Our findings indicate that some LLMs very closely approximate human expert judgments about the compliance (or lack thereof) of the excerpts in our dataset, such as Gemini 2.5 Pro (Comanici et al., 2025), which achieves a rank correlation of 0.856, as shown in Table 3.

In short, our contributions include:

- **Sample generation pipeline**: An open source repository for the LLM-based generation of plausible AIA technical documentation excerpts, which we use to generate the samples in AIReg-Bench, and which can be reused for other AI compliance evaluation and training initiatives.

- **Dataset**: The above pipeline is used to generate a distribution of samples that are then annotated by legal experts to create the AIReg-Bench open benchmark dataset, which can be used today to evaluate the effectiveness of LLMs at AIA compliance assessments — and which, in the future, is extensible to other AIR.

- **Experiments**: The first application of the benchmark to evaluate the performance of 10 frontier LLMs at the task of AIA compliance assessments.

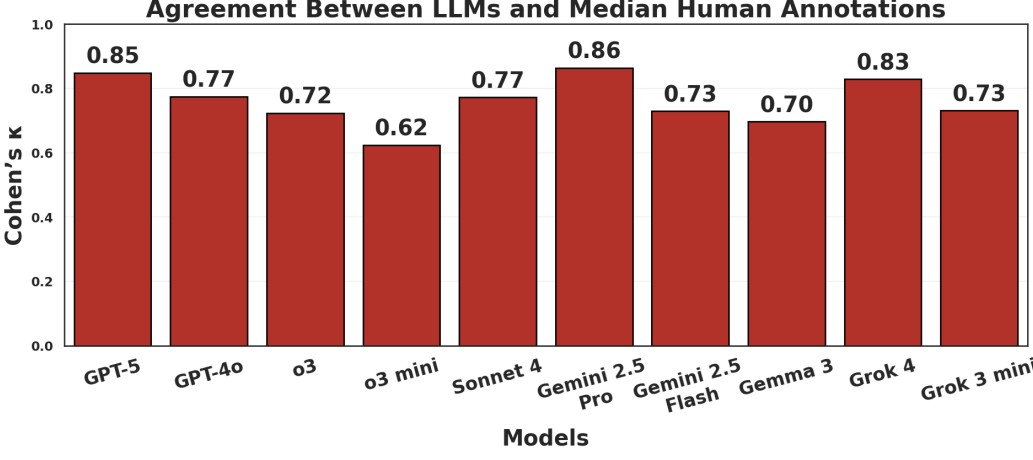

Figure 1: **Cohen's $\kappa$ (quadratically weighted) scores across frontier language models**, showing the level of agreement on compliance judgments (on a 1-5 Likert scale) between these models and the median legal expert in our team, taken over the entire AIReg-Bench dataset.

## 2 SAMPLE GENERATION PIPELINE (SOLUTION PART I)

Our first contribution is a sample generation pipeline which leverages an LLM to produce technical documentation excerpts, whose plausibility we have validated with legal experts (as described below in Section 2.2). This method, we argue, has standalone value, as it can be adapted to extend this benchmark or to generate new evaluation (and perhaps even training) datasets.

When conducting a compliance assessment, regulators, consultants, and internal audit teams may draw upon many different kinds of input, including technical documentation, source code, and transcripts of staff interviews. However, in designing our benchmark, we optimized for simplicity and ease of use by focusing on a single type of input. Specifically, since AI compliance assessors have identified technical documentation as "the most important factor" in assessing whether an AI system complies with the governing regulations (Li & Goel, 2025) — something that we validated in our own interviews with AIR compliance experts (described in Appendix D) — we decided to make technical documentation (or excerpts thereof) the sole type of input to inform compliance assessments in our benchmark.

The creation of our sample generation pipeline was motivated by two key bottlenecks: first, little to no AIA technical documentation of real AI systems is publicly available, perhaps due to the confidentiality or legal privilege surrounding such assessments (Guha et al., 2024); and second, paying experts to create them anew is prohibitively expensive (Pipitone & Alami, 2024). Therefore, inspired in part by Sovrano et al. (2025), who use an LLM to assist human drafting of technical documentation for AIA use cases, we designed a multi-stage pipeline, with gpt-4.1-mini (OpenAI, 2025) at its center, which generates plausible technical documentation excerpts efficiently and at scale. The stages of this pipeline are described in Section 2.1 and illustrated in Fig. 2. Although we also experimented with o3-mini (OpenAI, 2025b), 4o-mini (OpenAI, 2024), and gpt-5 (OpenAI, 2025a), gpt-4.1-mini was ultimately chosen for the pipeline due to its price point and the satisfactory outputs produced during our experimentation phase.

In devising this pipeline, we set forth several design criteria for the samples it generates. Many of these criteria, which are supplied in full in Appendix C, were written to ensure the benchmark remains manageably-scoped and easy to use. For example, we decided that our documentation should only depict HRAI systems within the scope of the AIA (EU, 2024, Art. 6). Although we could have chosen any subset of AIA requirements to treat as a proof of concept for this benchmarking effort, we felt the AIA's HRAI requirements were an especially worthwhile subject because many regard them as "the most important part of the AIA" (Araszkiewicz et al., 2022). We also decided that each sample should be written only from the perspective of an AI system provider attempting to demonstrate compliance with a single article within the AIA (EU, 2024, Art. 2).

Additionally, we labored to ensure our excerpts were realistic (i.e., representative of real-world technical documentation) and diverse (covering a range of AI systems with different intended uses and varying levels of compliance). As a concrete example, to give researchers control over the makeup and diversity of the distribution, we designed our pipeline to be steerable, allowing for the targeted generation of excerpts that are more or less likely to be compliant. The details of how this control was applied in AIReg-Bench are provided in Section 3, and further discussion of our design criteria and the rationale for these criteria can be found in Appendix C.

## 2.1 STAGES OF SAMPLE GENERATION

The design criteria described above were enforced through prompt engineering during each stage of technical documentation excerpt generation, as outlined below:

1. First, gpt-4.1-mini is prompted to generate high-level overviews of AI systems, which fall into several use cases, such as road traffic control and credit scoring. By design, these use cases should be classified as high-risk under the AIA (EU, 2024, Art. 6(2); Ann. III).

2. For each of these use cases, gpt-4.1-mini is given the system overview and a single AIA article (either Art. 9, 10, 12, 14, or 15) as context and then prompted to generate 'compliance profiles' (i.e., instructions for whether and in what way the AI system should breach a selected article) for each overview-article combination. Within these profiles is a short summary of how a selected AI system could breach a selected article.

3. For each of these compliance profiles, gpt-4.1-mini is prompted to generate an excerpt of technical documentation, using the relevant article and AI system overview as context.

The prompts used in each stage are included in Appendix E.

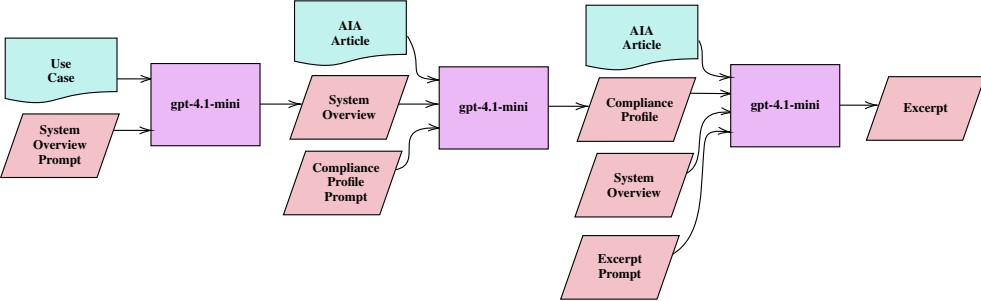

Figure 2: Illustration of the AIReg-Bench Technical Documentation Excerpt Generation Pipeline.

## 2.2 VALIDATION OF PLAUSIBILITY

To validate the plausibility of the pipeline's outputs, each excerpt in the AIReg-Bench dataset was reviewed by three legal experts (the same team of law school students, law graduates, and qualified lawyers, all regulation specialists, who supplied the annotations for the dataset, described in Section 3). As described in the annotation instructions (Appendix F), these legal experts were asked to label each excerpt with the probability that it is plausible — i.e, that it is realistic, logically consistent, and reflects the type of technical documentation that a Fortune 500 Europe AI provider might realistically hand over to its compliance assessor or internal audit team. These labels help assess whether the process of using an LLM to generate technical documentation succeeded.

To be more specific, for each excerpt, annotators were asked to provide a score on a 1–5 Likert scale (Likert, 1932), where 1 indicated a very low probability of plausibility and 5 indicated a very high probability of plausibility (the exact phrasing of the Likert scale given in Appendix F). To accompany these quantitative scores, annotators were asked to provide a qualitative free text justification of the Likert score (i.e., a free text plausibility analysis). These scores and text entries are available in the AIReg-Bench GitHub repo.

The median plausibility score provided by annotators was 4 (i.e., high probability of plausibility).

| | Traffic Safety (Use1) | Gas Delivery (Use2) | Education (Use3) | Exam Proctor. (Use4) | Job Hiring (Use5) | Job Term. (Use6) | Emergency (Use7) | Credit Scoring (Use8) | Total |
|---|---|---|---|---|---|---|---|---|---|
| Likert score 1 | 1 | 2 | 0 | 0 | 1 | 0 | 0 | 0 | 4 |
| Likert score 2 | 6 | 5 | 4 | 1 | 6 | 1 | 3 | 4 | 30 |
| Likert score 3 | 9 | 5 | 7 | 3 | 9 | 6 | 7 | 4 | 50 |
| Likert score 4 | 11 | 17 | 11 | 16 | 16 | 19 | 15 | 17 | 122 |
| Likert score 5 | 18 | 16 | 23 | 25 | 13 | 19 | 20 | 20 | 154 |
| All scores | 45 | 45 | 45 | 45 | 45 | 45 | 45 | 45 | 360 |

Table 1: AIReg-Bench dataset overview of human annotations for plausibility on a Likert scale.

## 3 ANNOTATIONS AND DATASET (SOLUTION PART II)

Our second contribution involves using the sample generation pipeline that we created to generate a balanced distribution of samples, which legal experts then annotated in order to create the AIReg-

Bench open benchmark dataset. This dataset is composed of 120 technical documentation excerpts that reflect 8 different use cases (intended uses) for AI systems and varying compliance profiles. A list of the use cases from the creation of AIReg-Bench is included as Appendix H and the prompt used to generate compliance profiles is included as Appendix E.2.

More specifically, to create a diverse distribution of samples, with some shaped to have more compliant properties and others less so, we programmed the generation pipeline to steer one third of samples towards compliance and the remainder towards non-compliance. However, ultimately, the 360 Likert scale compliance labels (3 per excerpt) provided by our annotators reflect the true diversity of our dataset, as shown in Table 2.

| | Traffic Safety (Use1) | Gas Delivery (Use2) | Education (Use3) | Exam Proctor. (Use4) | Job Hiring (Use5) | Job Term. (Use6) | Emergency (Use7) | Credit Scoring (Use8) | Total |
|---|---|---|---|---|---|---|---|---|---|
| Likert score 1 | 6 | 10 | 8 | 7 | 5 | 8 | 3 | 9 | 56 |
| Likert score 2 | 11 | 13 | 12 | 11 | 10 | 9 | 12 | 7 | 85 |
| Likert score 3 | 10 | 5 | 5 | 9 | 11 | 6 | 11 | 8 | 65 |
| Likert score 4 | 4 | 3 | 7 | 4 | 4 | 8 | 5 | 8 | 43 |
| Likert score 5 | 14 | 14 | 13 | 14 | 15 | 14 | 14 | 13 | 111 |
| All scores | 45 | 45 | 45 | 45 | 45 | 45 | 45 | 45 | 360 |

Table 2: AIReg-Bench dataset overview of human annotations for compliance on a Likert scale.

### 3.1 LEGAL EXPERT ANNOTATION OF THE EXCERPTS

Generating annotations in the legal domain often demands specialized legal expertise reflecting deep subject-matter knowledge (Guha et al., 2024). Accordingly, multiple legal natural language processing and LLM benchmarks have leveraged legal expert annotators (including law school students and lawyers) (Wang et al., 2025b; Zheng et al., 2025; Östling et al., 2024; Wang et al., 2023; Shen et al., 2022; Hendrycks et al., 2021b; Leivaditi et al., 2020; Zhong et al., 2019; Duan et al., 2019; Wilson et al., 2016). Following this pattern, we used a team of six legal experts (law graduates, law students, or qualified lawyers) to review and annotate the technical documentation excerpts in AIReg-Bench.[1] As detailed in Appendix I, each of these annotators listed regulation as one of their specializations. Prior to annotation, these legal experts attended a training session, led by a legally-trained co-author, dedicated to the AIA and its relevant articles. During annotation, labels were quality-checked by a law school graduate co-author.

In the end, each excerpt was reviewed by three legal experts. As described in the annotation instructions (Appendix F), for each excerpt, these annotators were asked to provide a score on a 1–5 Likert scale, where 1 indicated a very low probability of compliance with the relevant AIA article and 5 indicated a very high probability of compliance with that article (the exact phrasing of the Likert scale given in Appendix F).[2] To accompany these quantitative scores, annotators were asked to provide a qualitative free text justification of the Likert score (e.g., a free text compliance analysis). These

---

[1]There are very few potential annotators with expertise in the EU AI Act who possess both the willingness and the capacity to carry out extended annotation tasks. For this reason, we broadened the eligibility criteria to include annotators with legal training more generally— although, ultimately, our entire team listed regulation as one of their specializations (see Appendix I).

[2]Annotators also mark whenever they found assigning a Likert score "difficult". These markings can theoretically be used to segregate the more challenging samples. Notably, however, few annotators choose to use this demarcation in practice.

justifications can, in theory, be semantically compared to the justifications produced by LLMs to provide another way to measure LLM compliance assessment capabilities.

Appendix J contains a select annotation example. Appendix K contains an analysis of recurring themes in these annotations. The annotators' inter-rater reliability, as measured with Krippendorff's alpha coefficient (Krippendorff, 2018), was 0.651. This suggests a moderate level of agreement, though too low for drawing strong conclusions, likely reflecting the subjectivity of legal judgements.

Variance is heavily structured by two annotators with opposing biases (-0.917 and +0.600). Removing these annotators increases Krippendorff Alpha by +0.1343 to 0.786. That said, for the analyses in this paper, we retained all annotations to avoid post hoc exclusion.

The highest average disagreement occurred around Article 10 and Article 15, with mean standard deviations of 0.638 and 0.612 respectively. The most severe disagreement involved the intended use of Credit Scoring (Use 8) in light of Article 9, whose scores (1, 5, 5) resulted in the dataset's highest standard deviation (1.89).

## 4 EXPERIMENTS (PROVING OUR SOLUTIONS ADDRESS THE PROBLEM)

Our third contribution is the first application of the benchmark. We have created an evaluation of 10 frontier language models using the AIReg-Bench benchmark. This helps us understand, the current performance of frontier LLMs, out-of-the-box and without fine-tuning, at the AIA compliance assessment task.

In this evaluation, we prompted the LLMs to carry out the same task as the human annotators, supplying them with the identical documentation, system descriptions, and AIA articles that had been provided to the human annotators, along with instructions that were highly similar to those given to the human annotators (detailed in Appendix G). A key distinction, however, is that the annotators were free to consult external sources—such as websites or existing literature—whereas the LLMs were restricted to the materials explicitly supplied.

Each LLM generated annotations for all 120 excerpts, using the same format as the human expert annotators: Likert scale scores for compliance, accompanied by textual justifications for those scores. The LLM compliance scores were then compared with the median scores assigned by the human annotators, allowing us to evaluate how closely each model approximated human compliance judgments. The key statistics from this evaluation are included in Table 3.

| Model | $\kappa_w$ ($\uparrow$) | $\rho$ ($\uparrow$) | Bias ($\to 0$) | MAE ($\downarrow$) |
|---|---|---|---|---|
| GPT-5 (OpenAI, 2025a) | 0.849 | 0.838 | -0.067 | 0.450 |
| GPT-4o (OpenAI et al., 2024) | 0.775 | 0.842 | 0.458 | 0.558 |
| o3 (OpenAI, 2025c) | 0.723 | 0.809 | -0.192 | 0.658 |
| o3 mini (OpenAI, 2025b) | 0.624 | 0.798 | 0.742 | 0.775 |
| Claude Sonnet 4 (Anthropic, 2025) | 0.772 | 0.779 | -0.150 | 0.600 |
| Gemini 2.5 Pro (Comanici et al., 2025) | 0.863 | 0.856 | -0.225 | 0.458 |
| Gemini 2.5 Flash (Comanici et al., 2025) | 0.729 | 0.825 | -0.108 | 0.625 |
| Gemma 3 (Kamath et al., 2025) | 0.696 | 0.757 | 0.258 | 0.692 |
| Grok 4 (xAI, 2025b) | 0.829 | 0.829 | 0.242 | 0.475 |
| Grok 3 mini (xAI, 2025a) | 0.730 | 0.810 | 0.492 | 0.592 |

Table 3: **Agreement between LLMs and humans.** Columns report agreement between LLM and median human compliance scores across AIReg-Bench: quadratically weighted Cohen's $\kappa_w$; Spearman's $\rho$; Bias (mean signed difference, LLM−human); and MAE (mean absolute error).

All evaluated models demonstrated at least modest alignment with human expert judgments, with o3 mini showing the weakest Cohen's Kappa agreement (0.624). At the other end of the spectrum, Gemini 2.5 Pro achieved the highest level of agreement (0.863), as well as the best rank correlation (0.856) and mean absolute error (0.458). In fact, Gemini 2.5 Pro's compliance scores were within one point of the median human expert score for all but 7 out of 120 human expert median annotations (see Figure 3).

Despite prompts designed to mitigate sycophancy and acquiescence bias (Fanous et al., 2025), some models tended to assign higher compliance scores than human experts. Along this dimension, o3 mini and GPT-4o performed worst, with o3 mini strictly exceeding the median human expert score in 54.2% of excerpts, while only strictly falling below the median human expert's score in 1.7% of excerpts (see Table 4).

Ablations on GPT-4o revealed that modifying the prompt to request "harsh" and "critical" responses can reduce bias, but at the cost of declines across all three other metrics (see Table 5).

## 5 BACKGROUND AND RELATED WORK

In this section, we provide context for our work by reviewing some foundational concepts as well as the body of prior research has explored related methods and applications.

### 5.1 LEGAL AND OTHER LLM BENCHMARKS

Benchmark datasets that let researchers quantitatively measure how well an LLM performs a task have become an important factor in developing trust in these models (Guha et al., 2024). Although some popular benchmarks broadly assess LLM capabilities (Hendrycks et al., 2021a; Rajpurkar et al., 2016), where models are evaluated on specific tasks, it is desirable for benchmarks to be tailored more closely to those tasks (Peng et al., 2024). In this regard, a growing number of benchmarks have been developed to assess the performance of LLMs at legal tasks such as contract review (Hendrycks et al., 2021b; Wang et al., 2023), legal reading comprehension (Duan et al., 2019), and more (Zheng et al., 2025; Leivaditi et al., 2020). Guha et al. (2023) and Fei et al. (2023) both gather these prior benchmarks as well as other resources into aggregate LLM legal benchmarks.

### 5.2 LLMS FOR LEGAL, COMPLIANCE, AND AIR COMPLIANCE TASKS

Researchers have applied LLMs to a wide variety of legal tasks (Ma et al., 2024; Lai et al., 2024; Siino et al., 2025). This includes various compliance assessment tasks (Hassani, 2024; Bolton et al., 2025; Chen et al., 2024; Wang et al., 2025a), including AIR compliance assessments. For example, Makovec et al. (2024) input datasets, model cards, README files, or other AI project artifacts into a RAG-enhanced GPT-4 that accesses relevant portions of the AIA to predict the compliance level of the AI system depicted in the input. Davvetas et al. (2025) use a RAG-equipped LLM (mistral-small3.2) that takes, as input, certain features of an AI system (such as the type of AI system and the intended use) and outputs the risk-level of the AI system according to the AIA. Similarly, Kővári et al. (2025) use in-context learning and RAG to create a chatbot that can help users self-assess compliance with the AIA. Meanwhile, Li et al. (2025) test various LLMs' ability to act as a rudimentary AIA compliance checker by accepting a hypothetical AI system as context and predicting whether it is prohibited by, permitted by, or out of scope of the AIA. A series of interlinked studies by Nokia Bell Labs employ LLMs to support AI practitioners in AIR compliance subtasks such as populating impact assessment reports (Bogucka et al., 2024a; Herdel et al., 2024; Bogucka et al., 2024b). Sovrano et al. (2025) use an LLM to help human-drafted technical documentation align with the requirements of AIA Article 11.

### 5.3 THE EU AI ACT

The AIA went into force in August 2024 (Lomas, 2024) and sets forth harmonized requirements for AI systems and models placed on the market or put into service in the EU (EU, 2024, Art. 1-2). In laying out requirements for these AI systems, the AIA leverages a "risk-based" approach (Mahler, 2022), by which the exact requirements that apply to a system are a function of its perceived degree of risk. Here, the most demanding requirements are reserved for those AI systems deemed to be high-risk (EU, 2024, Art. 6). Such high-risk AI systems (HRAI) must satisfy a number of requirements (EU, 2024, Chap. III, Sec. 2). Among those, the requirements that we have made the focus of our benchmark dataset relate to risk management systems, data and data governance, record keeping, human oversight, as well as accuracy, robustness, and cybersecurity (EU, 2024, Art. 9, 10, 12, 14, 15).

### 5.4 NON-LLM ALGORITHMS THAT ASSESS AIR COMPLIANCE

It is important to distinguish the works in Section 5.2 as well as this work from those works that use algorithms other than LLMs, including but not limited to benchmark suites, to evaluate whether AI systems, including LLM-based systems, to comply with AI regulations (Prandi et al., 2025; Marino et al., 2024; Guldimann et al., 2024; Walke et al., 2023). While these works potentially present interesting accompaniments to the LLM-based approaches that this work aims to benchmark, this work does not seek to create a benchmark for these other methods.

## 6 DISCUSSION

Here, we consider some of the limitations of our work, anticipate some of the questions that the research community might reasonably have about our methods, and describe how we addressed those.

### 6.1 CHALLENGES OF LEGAL BENCHMARKING

**Compliance assessments are subjective.** Complicating legal benchmarking is the fact that legal tasks often involve subjective judgments (Guha et al., 2024; Ma et al., 2023). Compliance, in particular, has been described as "not binary" (Wu & van Rooij, 2021). This was one motivation for our use of Likert scale annotations keyed to the "probability" of compliance rather than binary labels of "compliant" or "non-compliant." The Likert scale is viewed as a reliable way to "transform subjective qualitative data into quantifiable metrics" (Koo & Yang, 2025) and has previously been used for benchmarking of LLMs in subjective realms (Bojić et al., 2025).

Potentially increasing the subjectivity of legal annotations is the nascency of the AIA, which lacks the established guidelines and court rulings that typically help annotators reach more consistent conclusions (Goodman, 2023). To quantify this subjectivity, we measured annotators' inter-rater reliability via the Krippendorff's alpha coefficient (Krippendorff, 2018), which came to 0.651. This reflects moderate agreement between annotators, albeit lower than the levels of inter-rater reliability typically expected in domains with less subjective tasks.

To mitigate the variance that arises from the subjectivity of compliance assessments, each excerpt was scored independently by three of our six annotators, and most analyses were conducted using the median of these scores. Additionally, to pinpoint areas of greater subjectivity, we asked annotators to flag compliance annotations that were more "difficult." However, in practice, few annotations were flagged as challenging, potentially as annotators struggled to identify which cases were more challenging (Rother et al., 2021).

**Compliance assessments are a moving target.** It has been said that compliance assessments require clear and specific guidelines, including relevant case law (Kilian et al., 2025; Schuett, 2024). But, in its present state, the AIA arguably lacks these. The text of the law has not been interpreted by courts (Yew et al., 2025). The obligations outlined in the AIA have yet to be clarified by accompanying technical standards (European Committee for Electrotechnical Standardization, 2024; European Commission, 2022). They are also subject to ongoing amendments (EU, 2024, Art. 96). Accordingly, AIReg-Bench should be viewed as a snapshot of AIA conformity assessments in September 2025. It does not and cannot reflect developments occurring after this date.

### 6.2 CHALLENGES OF LLM-DRIVEN SAMPLE GENERATION

Benchmark dataset samples should be representative of real-world data (Sourlos et al., 2024). Broadly speaking, there is evidence that LLMs (especially larger ones) can effectively produce synthetic samples satisfying this criteria (Maheshwari et al., 2024). More directly relevant here, it has been shown that LLMs can effectively generate (or improve) legal documents (Su et al., 2025; Lin & Cheng, 2024; Hemrajani, 2025; Gray et al., 2025), technical specifications (Xie et al., 2025), compliance documentation (Wang et al., 2025d; Hassani, 2024; Kumar & Roussinov, 2024), and even the type of technical documentation required under the AIA (Sovrano et al., 2025).

However, some are sceptical about the ability of LLMs to generate realistic outputs in these domains (Posner & Saran, 2025; Roberts; Shen et al., 2022). Critics point out that LLMs often lack domain-

specific tacit knowledge, have difficulty maintaining coherent reasoning across extended contexts, and may hallucinate facts or references (Rasiah et al., 2024; Huang et al., 2024; Dahl et al., 2024; Magesh et al., 2025). Regarding legal tasks specifically, critics note how LLMs' struggle to interpret legal terminology, grasp case context, and execute complex analyses, potentially resulting in errors (Wang et al., 2024; Roberts; Shen et al., 2022). With this in mind, a number of guardrails were put in place to enforce plausibility in our dataset samples and to create transparency around whether and where those guardrails fell short. For example:

- The sample generation method was informed by the series of interviews with actual compliance assessment experts, described in Appendix D. We also consulted similar interviews performed by Li & Goel (2025) and the recommended protocols for manual compliance assessments for the AIA (Floridi et al., 2022; Thelisson & Verma, 2024; Lillo Campoy et al., 2024; Palumbo et al., 2025) and other AIR (The Institute of Internal Auditors, 2023; National Institute of Standards and Technology, 2023; Brogle et al., 2025).

- The sample generation method was co-engineered by a law school graduate co-author who has been involved in the drafting of the codes of practice accompanying the AIA.

- Before the samples in AIReg-Bench were generated, the method was subjected to an iterative, plausibility-focused development process with a subset of our expert annotators (including an EU qualified lawyer). This iterative process significantly improved our prompts (as measured by the plausibility of the excerpts they generated) compared to those produced using small prompts, which we initially considered. As an example, unlike those in our final excerpts, the compliance violations generated by small prompts were exceedingly obvious, superficial, and unrealistic — a limitation also noted by Nguyen et al. (2025).

- During annotation, our legal expert annotators scored AIReg-Bench samples for plausibility, with those scores and their accompanying text explanations being made public, in their entirety, as part of the AIReg-Bench dataset.

## 6.3 CHALLENGES OF LLM-DRIVEN COMPLIANCE ANALYSES

Beyond the challenges of generating realistic documentation samples with LLMs, there may also be hurdles to using LLMs to perform legal analyses on text. Although some research efforts have found that LLMs match or exceed human accuracy when performing such analyses (Martin et al., 2024), others have questioned whether LLMs can perform this task effectively (Buckland, 2023; Doyle & Tucker, 2024; Li et al., 2024; Network, 2025; Mik, 2024). While the evidence presented in this paper is not definitive, we hope that our benchmark offers an initial step towards clarifying how well LLMs perform at compliance assessments, both in comparison to human experts and to one another. Our hope is that this will inspire other efforts to quantitatively evaluate the performance of LLMs at AIR compliance analyses and other legal tasks.

## 6.4 RISKS OF LLM-DRIVEN COMPLIANCE ANALYSES

As discussed in Section 1, there are potential benefits to LLM-driven AIR compliance analyses. However, it is important to point out that they could also carry risks. For example, given LLMs' ongoing tendency to make errors, some argue that there may be dire consequences when lawyers place "blind faith in an LLM" (Moriarty, 2023) and that it may even represent a violation of professional ethics (Browning, 2024). Conversely, the developers of AI systems could try to "game" these LLMs, manipulating their technical documentation so as to achieve desirable compliance assessment outcomes. To help inform the conversation about these risks~~this conversation~~, and avoid generalization within that conversation, we believe it is invaluable to quantitatively evaluate the performance of LLMs *for the task at hand*, comparing their performance to human performance at that task in an evidence-based manner. Hence this benchmark.

## 7 FUTURE WORK

This benchmark should serve as a starting point for tracking LLM progress at AIR compliance assessments, rather than as a finish line indicating readiness for deployment in legal practice. Mov-

ing towards that goal will require additional benchmarks that build upon and extend AIReg-Bench. Some suggested directions for extensions to AIReg-Bench are outlined below.

**Extension to other AIR.** AIReg-Bench is currently scoped to a subset of the AIA's requirements for HRAI systems. In the future, it would be natural to extend AIReg-Bench to the rest of the AIA's requirements for HRAI systems as well as to its requirements for general purpose AI models (European Parliament, 2024, Chap. V). When other AIRs achieve the AIA level of maturity, AIReg-Bench could also be extended to cover those AIR as well. In all cases, we believe that AIReg-Bench's overall playbook could be re-used, though the excerpt generation pipeline would need to be subtly reconfigured for these regulations and the excerpts annotated in light of the different compliance requirements.

**Extension to further LLM-powered annotators.** It would be valuable to extend our benchmarking in Section 4 to include more models: including fine-tuned legal LLMs, which some argue perform better at legal tasks (Fei et al., 2023; Dominguez-Olmedo et al., 2024), as well as LLMs with tool-use (e.g., incorporating RAG or web search) (Makovec et al., 2024; Davvetas et al., 2025; Wang & Yuan, 2025). Though, while to our knowledge, no such models exist yet, there would also be value in fine-tuning LLMs for AIA or AIR compliance and then evaluating them using AIReg-Bench.

**Extension to real-world documentation.** AIReg-Bench consists of LLM-generated excerpts of technical documentation for the AIA. We relied on these LLM-generated excerpts, whose plausibility we manually verified, since public access to ~~authentic~~real technical documentation from real AI developers is limited, perhaps due to the confidentiality, legal privilege, or the relative nascency of the AIA. That said, we recognize the value, specifically as it relates to construct validity, of a benchmark built from real instances of AIA technical documentation. We therefore encourage any actors with access to such documentation to consider publishing (an anonymized version of) itand, furthermore, to contact us if we can assist with that process.

**Extension to AIA technical standards.** Like other EU product regulation, the AIA will utilize harmonized standards, i.e., "more concrete" (Siegmann & Anderljung, 2022) technical specifications prepared by the EU's external standardization organizations (CEN, CENELEC and ETSI). Compliance with these specifications, which are still under development (European Committee for Electrotechnical Standardization, 2024; European Commission, 2022), will "have the legal effect of establishing a presumption of conformity" with the AIA (Mazzini & Scalzo, 2023). Once issued, it would be important to pass these specifications, as additional context, into any AIA compliance assessment algorithm.

**Extension to multi-turn interactions.** Our interviews of AIR compliance professionals suggested that, in practice, compliance assessments often involve a long and complex dialogue between legal teams, technical staff, and regulators (see Appendix D). AIReg-Bench condenses the entire compliance assessment process into a single-turn interaction, based on a fixed set of synthetic artifacts. Such scoping and simplifications are common in benchmarking, though many scholar stress the need to shift towards more interactive modes of evaluating AI system capabilities (Ibrahim et al., 2024; Eriksson et al., 2025). Future benchmarks could build on AIReg-Bench by evaluating LLMs' multi-turn ability to collaborate with human teams and contribute to complex legal dialogues (Kővári et al., 2025), rather than merely producing one-shot assessments. Similarly, future benchmarks might explore scenarios where LLMs are merely used as the starting point in a multi-turn, human-in-the-loop compliance assessment process.

## 8 CONCLUSION

In this work, we introduced AIReg-Bench, an open benchmark designed to quantitatively evaluate the performance of LLMs on AIA compliance assessments. By combining an LLM-driven sample generation pipeline with expert legal annotations, AIReg-Bench provides a scalable, realistic, and extensible foundation for assessing how closely models align with human expert compliance judgments. Our initial experiments with frontier LLMs demonstrate both the promise and current limitations of these systems in performing this task. While AIReg-Bench is only an initial step, we hope it catalyzes further research into LLM-driven AIR compliance assessments.

## 9 ETHICS STATEMENT

We do not believe that our paper submission raises questions regarding the Code of Ethics. All legal expert annotators are co-authors on the work. What is more, since this work ultimately strives to benchmark and improve LLM-driven compliance assessments, it should ultimately serve to increase compliance with the AI regulations that tend to encode important societal values such as fairness and safety.

## 10 REPRODUCIBILITY STATEMENT

Here, we discuss the efforts that have been made to ensure reproducibility. The parts of the main paper, appendix, and supplemental materials (including the project GitHub repository) that will help with reproducibility are as follows:

**Sample generation pipeline**: The full code for the Sample generation pipeline is available at the AIReg-Bench GitHub repository https://anonymous.4open.science/r/aireg-bench-5259/ . In addition, the following are included here as appendices:

- Sample generation pipeline prompts (Appendix E)
- Use cases for the AIReg-Bench sample generation pipeline (Appendix H)

**AIReg-Bench dataset**: The full AIReg-Bench dataset, including samples and annotations, is available at the AIReg-Bench GitHub repository https://anonymous.4open.science/r/aireg-bench-5259/ In addition, the following are included here as appendices:

- Legal expert annotation instructions (Appendix F)

**Evaluation of frontier LLMs using AIReg-Bench**: The full code for the evaluation is available at the AIReg-Bench GitHub repository https://anonymous.4open.science/r/aireg-bench-5259/

. In addition, the following are included as appendices:

- LLM annotation instructions (Appendix G)

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

## A  LLM USAGE IN THIS WORK

LLMs were used in this work as follows:

- LLMs were used to aid or polish writing. Specifically, they were used to identify errors or weaknesses in writing, to generate an initial draft of the Conclusion (Section 8), and to generate bibtex.

- LLMs were used for retrieval and discovery. Specifically, they were used for finding related work.

- LLMs were used for other purposes, such as the generation of the samples (described in Section 2), tables, and streamlining code development.

## B ADDITIONAL FIGURES

This section presents supplementary figures and tables that, for the sake of brevity, have been omitted from the main body of this paper.

Figure 3 (left) shows confusion matrices comparing the median human expert compliance scores (rows) with the LLM scores (columns). Darker cells indicate more frequent score combinations. The top matrix is for Gemini 2.5 Pro, the best-performing model across most evaluated metrics. The majority of score combinations sit along the diagonal (72/120), showing strong agreement between Gemini 2.5 Pro and the median human expert annotator. The bottom matrix averages the frequency of score pairings over all evaluated models, and here, the distribution is more diffuse.

Figure 3 (right) reports the mean absolute error (MAE) between LLM compliance scores and median human expert scores, broken down by use case (columns) and article (rows). Darker cells reflect larger errors. The top heatmap shows the MAE breakdown for Gemini 2.5 Pro, while the bottom heatmap averages MAE across all evaluated models. Notably, in the bottom heatmap only three cells exceed an MAE of 1.0, indicating a consistently strong average agreement between LLMs and human experts.

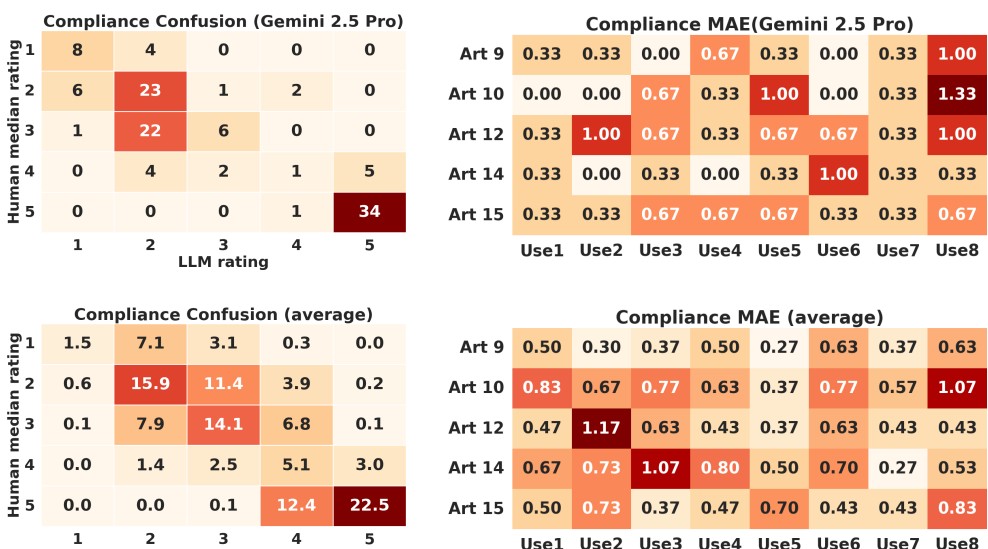

Figure 3: **Heatmaps of compliance performance.** The left panels show the distribution of compliance ratings (in 'confusion matrix'), comparing the median human expert with LLMs. The right panels show mean absolute error (MAE) across use cases and articles. Results are shown for Gemini 2.5 Pro (top) and as an average over all evaluated LLMs (bottom). Use1-8 reflect the intended uses from Table 2.

Figure 4 plots model cost (x-axis, output price per M tokens) against compliance agreement with human expert ratings (y-axis, Cohen's $\kappa$ weighted quadratically).[3] Each point on the graph corre-

---

[3] Gemma 3 is not included as it is not available via paid API access.

sponds to an evaluated LLM and those highlighted in red (Gemini 2.5 Pro and Grok 3 mini) lie on the Pareto frontier meaning that no model is both cheaper and more compliant.

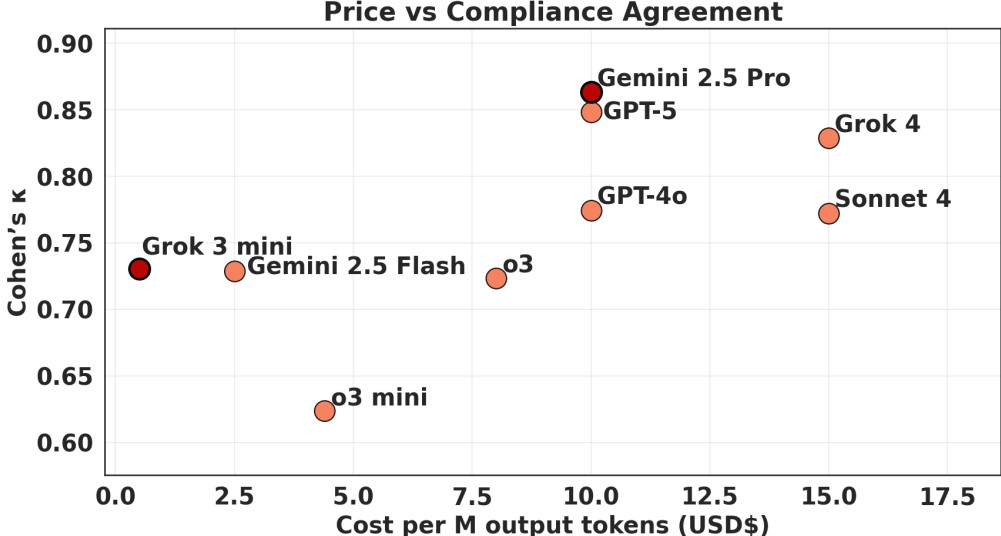

Figure 4: **Pareto frontier of model cost versus compliance agreement (Cohen's $\kappa$).** Each point represents a model, plotted by price (x-axis) and agreement with human expert ratings (y-axis). Pareto-efficient models are shown with red markers. Labels denote model names. We were not able to capture the cost of our particular implementation of Gemma, as API use is free but has a low restriction on output tokens.

Table 4 presents the accuracy of LLMs in replicating human expert scores exactly and, where mismatches occur, the direction of error. Despite prompts designed to mitigate sycophancy and acquiescence bias, some models tended to assign higher compliance scores than human experts. Two of the worst models in this regard were o3 mini and GPT-4o, with o3 mini strictly exceeding the median human expert score in 54.2% of excerpts, while only strictly falling below in 1.7%. The model least prone to over-estimating the compliance level of excerpts was Gemini 2.5 Pro, which was also the model whose scores exactly matched those of the median human expert most frequently (for 60% of excerpts). Gemini 2.5 Flash achieved the best F1 score (0.913), meaning it excelled at correctly flagging compliant cases (score 4-5) and avoiding mislabelling non-compliant ones (score 1-3).

| Model | % Exact (↑) | % Over (↓) | % Under (↓) | F1 (↑) |
|---|---|---|---|---|
| GPT-5 | 57.5 | 16.7 | 25.8 | 0.903 |
| GPT-4o | 52.5 | 42.5 | 5.0 | 0.846 |
| o3 | 38.3 | 20.8 | 40.8 | 0.903 |
| o3 mini | 44.2 | 54.2 | 1.7 | 0.736 |
| Sonnet 4 | 46.7 | 20.0 | 33.3 | 0.845 |
| Gemini 2.5 Pro | 60.0 | 10.0 | 30.0 | 0.911 |
| Gemini 2.5 Flash | 41.7 | 22.5 | 35.8 | 0.913 |
| Gemma 3 | 40.0 | 39.2 | 20.8 | 0.796 |
| Grok 4 | 58.3 | 30.8 | 10.8 | 0.860 |
| Grok 3 mini | 53.3 | 42.5 | 4.2 | 0.830 |

Table 4: **Compliance Likert score differences between LLMs and human experts.** Columns report accuracy of LLMs across AIReg-Bench, including the percentage of exact matches, over-estimates, and under-estimates relative to the median human expert, as well as the F1 score from binary classification (scores 1–3 vs. 4–5).

Table 5 presents ablation results for GPT-4o, comparing the baseline model to three ablated versions: one without a tone prompt (see Appendix G), one with a harsh tone prompt, and one without access to the relevant AI Act text. The baseline tone prompt ("Your scores for both compliance and plausibility should be well-calibrated and objective. They should be rigorous but fair.") achieves the best performance with respect to Cohen's $\kappa_w$, Spearman's $\rho$, and MAE.

The harsh tone prompt ("Your scores for both compliance and plausibility should be critical. They should be harsh but fair.") reduces bias, but at the cost of declines across all three other metrics. When access to the text of the articles in the EU AI Act is removed, all performance metrics drop substantially, with GPT-4o's Cohen's $\kappa_w$ falling to 0.654, just above o3 mini's performance when provided with the text.

| Model | $\kappa_w$ ($\uparrow$) | $\rho$ ($\uparrow$) | Bias ($\rightarrow 0$) | MAE ($\downarrow$) |
|---|---|---|---|---|
| GPT-4o baseline | 0.775 | 0.842 | 0.458 | 0.558 |
| ablation (none) | 0.759 | 0.842 | 0.492 | 0.575 |
| ablation (harsh) | 0.722 | 0.791 | 0.125 | 0.642 |
| ablation (w/o articles) | 0.654 | 0.752 | 0.583 | 0.717 |

Table 5: **Ablation analysis of GPT-4o.** Ablations include removing or altering the prompt modifier (to be harsher), or withholding access to the AI Act text. Columns report agreement between LLM and median human expert scores across AIReg-Bench: quadratically weighted Cohen's $\kappa_w$; Spearman's $\rho$; Bias (mean signed difference, LLM−human); and MAE (mean absolute error).

Table 6 includes the results for well-known open-source language models fine-tuned for legal tasks: LLM Saul-7B-Instruct (Colombo et al., 2024b) and Saul-54B-Instruct (Colombo et al., 2024a). Both models are relatively small in size and, even with the potential advantages of fine-tuning on legal material, they underperform the weakest general-purpose frontier model in our evaluation with respect to Cohen's $\kappa_w$, achieving 0.183 and 0.596 respectively, compared to o3 mini's 0.624.

That said, the stark improvement from Saul-7B-Instruct to Saul-54B-Instruct is significant, highlighting the benefits of scaling to larger models. Notably, Saul-7B-Instruct struggles to consistently format its outputs as requested, a limitation observed in smaller language models more generally (Wang et al., 2025c). Many of its answers therefore had to be resampled until they were formatted appropriately.

| Model | $\kappa_w$ ($\uparrow$) | $\rho$ ($\uparrow$) | Bias ($\rightarrow 0$) | MAE ($\downarrow$) |
|---|---|---|---|---|
| Saul-7B-Instruct | 0.183 | 0.311 | 0.550 | 1.167 |
| Saul-54B-Instruct | 0.596 | 0.813 | 0.792 | 0.825 |

Table 6: **Experiments with Saul**: Columns report agreement between LLM Saul-7B-Instruct (Colombo et al., 2024b) and Saul-54B-Instruct (Colombo et al., 2024a) and median human compliance scores across AIReg-Bench: quadratically weighted Cohen's $\kappa_w$; Spearman's $\rho$; Bias (mean signed difference, LLM−human); and MAE (mean absolute error). Outputs from Saul-7B-Instruct were resampled until they were provided in a parsable format.

Table 8 summarizes the results of the alternative annotator test proposed by Calderon et al. (2025). The goal is to check whether replacing a human annotator with a given model would preserve the overall decisions. The winning rate captures the fraction of humans for whom the model would serve as an adequate substitute, while the average advantage probability reflects how often the model's annotation is at least as good as a human's across items. In our discrete label setup, a model wins on an item if its label is closer to the remaining annotators' labels than the human's.

| | GPT-5 | GPT-4o | o3 | o3 mini | Sonnet 4 | Gemini 2.5 Pro | Gemini 2.5 Flash | Gemma 3 | Grok 4 | Grok 3 mini |
|---|---|---|---|---|---|---|---|---|---|---|
| GPT-5 | – | | | | | | | | | |
| GPT-4o | 0.73 | – | | | | | | | | |
| o3 | 0.70 | 0.57 | – | | | | | | | |
| o3 mini | 0.55 | 0.87 | 0.45 | – | | | | | | |
| Sonnet 4 | 0.84 | 0.75 | 0.74 | 0.55 | – | | | | | |
| Gemini 2.5 Pro | 0.84 | 0.70 | 0.71 | 0.56 | 0.85 | – | | | | |
| Gemini 2.5 Flash | 0.73 | 0.67 | 0.75 | 0.51 | 0.77 | 0.74 | – | | | |
| Gemma 3 | 0.70 | 0.73 | 0.55 | 0.69 | 0.62 | 0.64 | 0.64 | – | | |
| Grok 4 | 0.85 | 0.84 | 0.59 | 0.65 | 0.86 | 0.86 | 0.72 | 0.61 | – | |
| Grok 3 mini | 0.68 | 0.92 | 0.55 | 0.77 | 0.78 | 0.67 | 0.70 | 0.78 | 0.82 | – |

Table 7: **Pairwise Cohen's Kappa scores**. This table shows the results of performing pairwise Cohen's Kappa scores of the models in our evaluation.

| Model | Items | Annotators | Winning Rate | Average Advantage Probability |
|---|---|---|---|---|
| GPT-5 | 60 | 3 | 0.6667 | 0.8500 |
| GPT-4o | 60 | 3 | 0.0000 | 0.7944 |
| o3 | 60 | 3 | 0.0000 | 0.6556 |
| o3 mini | 60 | 3 | 0.0000 | 0.7556 |
| Sonnet 4 | 60 | 3 | 0.0000 | 0.7333 |
| Gemini 2.5 Pro | 60 | 3 | 1.0000 | 0.9111 |
| Gemini 2.5 Flash | 60 | 3 | 0.0000 | 0.6722 |
| Gemma 3 | 60 | 3 | 0.0000 | 0.6722 |
| Grok 4 | 60 | 3 | 0.6667 | 0.8444 |
| Grok 3 mini | 60 | 3 | 0.0000 | 0.7500 |

Table 8: **Alternative Annotator Test**. This table shows the results of the test proposed by Calderon et al. (2025), which answers the question: if we replaced human annotators with another LLM, would the resulting labels stay the same? Here, winning rate is defined as the fraction of human annotators for whom the LLM passes the test (i.e., the model's annotation is closer to the other human scores than the alternative annotation). Meanwhile, average advantage probability is defined as the average probability that the LLM's annotation is at least as good as a human experts's on each item. Our implementation reproduces the paper's procedure as-is.

## C    DESIGN CRITERIA

The design criteria for AIReg-Bench were as follows:

- In keeping with good benchmark dataset practices (Sourlos et al., 2024), the samples should be representative of real-world technical documentation used in AIA compliance assessments. Since AIReg-Bench is intended for evaluating LLMs' ability to perform AIR compliance assessments, we sought to replicate as closely as possible the documentation that a human compliance assessor would consult during this process.

- The samples should only depict HRAI systems (EU, 2024, Art. 6) that are within the scope of the AIA (but not within the scope of any of its prohibitions or exceptions) and should

be drafted as if created by the AI system's provider (i.e., developer) (EU, 2024, Art. 2). These design criteria, we argue, add a degree of realism to the dataset, since providers of AI systems outside of these boundaries are less well incentivized to create detailed technical documentation. Moreover, by focusing solely on in-scope HRAI systems, it ensures that every document we generate is densely packed with compliance-critical details, many of which may not be required for assessing lower-risk systems.

- Aside from a high-level overview of the AI system, each sample's contents should be constrained to specific AIA requirements for HRAI systems. This ensures the benchmark tests models on fine-grained compliance analysis rather than on their ability to interpret overly broad or generic descriptions.

- In order to achieve a diverse distribution, the samples should be able to reflect a variety of use cases (i.e., intended uses) as well as different compliance scenarios (either compliant or non-compliant with relevant Articles of the AIA). Collectively, they should cover a variety of intended uses and compliance profiles: that is, some systems are compliant with the AIA, while others are not — and, in the case of the latter, the reasons for non-compliance vary.

## D  COMPLIANCE EXPERT INTERVIEWS

Since AIReg-Bench is intended for evaluating LLMs' ability to perform AIR compliance assessments, we sought to replicate as closely as possible the technical documentation that a human compliance assessor would consult during this process. To better understand the structure and contents of this particular documentation, we interviewed six compliance experts (distinct from our six annotators), asking them to provide details about the materials they consult (or would expect to be consulted) during AIR compliance assessments, including but not limited to those mandated by the AIA (EU, 2024, Art. 43).

Perhaps owing to the new and evolving nature of AIR compliance assessments, the consensus among interviewees was that there are still no universal standards for the materials to be consulted during this process. That said, the materials that were most commonly referenced by interviewees were summaries of an AI system's attributes and its development process — including, but not limited to model cards, data cards, descriptions of data preparation, training and red-teaming processes, and descriptions of governance or guardrailing measures.

Some of our experts suggested that these materials might be curated in preparation for a compliance assessment using business records and auditee interviews, and that several such materials may be integrated into a single instance of technical documentation. These interviewees indicated that, in practice, one such technical document can serve as the primary artifact in compliance assessments, even though compliance assessors may draw on a wider range of materials through iterative dialogue. To reflect the central role of technical documentation and to ensure our benchmark is simple to use, we represent each AI system with a single integrated technical document, rather than many such materials.

Although interviewees consistently highlighted a lack of clear standards for compliance assessments, many regarded the provisions and annexes of the AIA related to technical documentation as among the clearest and most detailed guidance for AIR assessments. Accordingly, our dataset is predominantly built around this regulation and, in particular, Annex IV and Chapter III, Section 2 of the Act (EU, 2024, Ann. IV, Chap. III(2)) — which we found to be most relevant when producing technical documentation for compliance assessments. By focusing almost-entirely on just these two parts of the Act and omitting its less relevant provisions or any ancillary requirements (such as harmonized standards), our technical documentation remains manageable in length, concentrating exclusively on the core requirements for a compliance assessment.

## E  SAMPLE GENERATION PIPELINE PROMPTS

Listed below are the prompts that were fed to gpt-4.1-mini during the sample generation pipeline, as well as the annotation instructions given to humans and LLM. Additional line breaks have been added for readability.

## E.1 SYSTEM OVERVIEW PROMPT

Your task is to generate four distinct AI system descriptions for the provided intended use.

Each AI system must employ only one or two domain-appropriate types of machine learning models or algorithms. You should pick the algorithm you feel is most appropriate for the use case in the contemporary era, but here are some examples of the types of algorithms that you might choose: MLP, CNN, Transformers (encoder-only, decoder-only, or encoder-decoder), SVM, RNN, Naive Bayes, GNN, Random Forest, KNN, GBDT, Linear Regression, transformer-based Large Language Model (LLM), transformer-based Vision Language Model (VLM), diffusion-based text-to-image generation model, or similar. Transformers can be used in distinct ways, including for processing different data types such as tabular data, text, audio, API calls, and more.

Your choices of models or algorithms should reflect those likely to be deployed in 2025. You should focus on realism given the particular application as well as domain-appropriateness. Systems must not employ biometric technologies or violate Article 5 of the EU AI Act.

Start each description with: "System 1." "System 2." "System 3." "System 4." Each description should be a single continuous paragraph. There should be an empty line separating each system description from the next.

*****Intended use*****
{*Insert intended use*}

For each system, provide a concise description consisting of the following four components, each in a single sentence:

[a] System Name and Type: State the AI system's name and the machine learning models or algorithms it relies on. When naming these AI systems, ensure the names are diverse, realistic, and professional (while also fictional). System names should be formed by using multiple, separate words; never form system names by concatenating words (e.g., "EducationSmart") or using portmanteaus (e.g.,"EduBoost").

[b] System use: Provide the system's intended usage and the specific sector or area it will serve.

[c] Objective and Technological Capabilities: Describe the system's primary objective and outline the technological capabilities that enable the achievement of this objective.

[d] Provider: Identify the natural or legal person, public authority, agency or other body that develops the system or that has the system developed and places it on the market or puts the system into service under its own name or trademark, whether for payment or free of charge. Where providers are companies, ensure their names are realistic and professional (while fictional); please do not simply concatenate words (e.g., "EducationSmart") or use portmanteaus (e.g.,"EduBoost").

[e] The AI subject: Define the persons (or groups of persons) who are directly affected by the AI system, experiencing its outcomes and consequences.

Return nothing else outside those four system descriptions, formatted in the manner outlined above.

## E.2 VIOLATION PROMPT

*****Task*****
You are acting as an EU AI Act compliance expert and machine-learning specialist. You will be given an extract from the EU AI Act and a short description of an AI system. Your task is to generate three distinct ways that the AI system could violate the provided extract.

You must not produce:

- violations that reflect errors in the technical documentation rather than the system (e.g., lies or missing details).
- violations without any justification, evidence, or granular details (e.g., "the model is biased" without elaboration).

You must produce:

- violations that are plausible and realistic

- violations that are subtle and not blatant

Start each example with: "Example 1.", "Example 2.", "Example 3." Each description should consist of four parts [a]-[d], each given 1-3 sentences. There should be an empty line separating each part from the next and each example from the next.

Example X:
[a] **Quotation:** <verbatim quote from the EU AI Act extract that will be violated>
[b] **Guideline:** <granular and realistic standards that experts would use to ensure compliance with the quotation>
[c] **Violation:** <precise account of a violation of guideline [b] that would imply non-compliance with extract [a]>
[d] **Justification:** <an explanation for why violation [c] breaches the quoted requirement and why it is realistic yet subtle>

Return nothing else outside those three examples, formatted in the manner outlined above.

*****Extract*****
{*Insert article text from EU AI Act*}

*****System*****
{*Insert system outline*}

### E.3    DOCUMENTATION PROMPT

*****Task*****
Please write the documentation so that it aligns with the compliance profile (but show it, don't tell it).

*****Compliance profile*****
{*Insert compliance profile*}

*****Details*****
You are acting as a compliance expert and machine-learning specialist. You are tasked with contributing to the writing of technical documentation for an AI system. You represent the provider of an AI system. You do not represent the system deployers and are not aware of their identities, though you need not explicitly reference this in your response.

To inform your response, you will be given an extract from the EU AI Act, a short description of an AI system, and a compliance profile. If the extract references provisions outside of the extract itself (whether from elsewhere in the EU AI Act or from external legislation) interpret them using the context in which they are referenced and your prior knowledge of the EU AI Act. You will produce a section of technical documentation (intended to inform a compliance assessment against the provided extract of the EU AI Act) for the specified AI system and its compliance profile.

Substance - You must (unless told otherwise by the compliance profile):

- Present all necessary provider decisions (with associated evidence and rationale) to facilitate a sober and detail-oriented compliance assessment.

- Discuss realistic system components and modalities representative of those typically used in 2025, reflecting current industry standards.

- Discuss realistic compliance measures representative of those typically used in 2025, reflecting current industry standards.

- Ensure the documentation is consistent with the provided system description and compliance profile.

- Ensure the documentation is internally consistent (e.g., system attributes and compliance measures fit together without contradiction, describing a coherent and realistic set of technical and operational facts).

- Ensure the system description contains a substantive, rather than a cursory, set of facts. To do so, you may need to fictionalise evidence, research, findings, and details to support your claims.

- Ensure any fictionalised numerical details and supporting evidence (e.g., dataset size, performance, benchmarks, adversarial testing, data processing) are realistic.

- Ensure that any quoted numbers are consistent with each other and can be plausibly combined (e.g., a model trained on a large number of samples would require a large amount of compute).

- Address only the provided extract of the EU AI Act; do not address other articles or related regulations.

- Ensure the system is not prohibited by Article 5 of the EU AI Act and is also not biometric.

Formatting - You must (unless told otherwise by the compliance profile):

- Begin your response with: **Article X**, where X is the number of the article given in the extract.

- Create subtitles for the different parts of your response that are appropriate for legal prose; avoid just repeating the provisions from the extract as subtitles.

- Tailor paragraph length and detail; each bullet should be addressed fully, typically in 150-300 words.

Style - You must (unless told otherwise by the compliance profile):

- Produce a professional and realistic simulation of the structured prose an auditor may receive.

- Use technical but accessible language, briefly clarifying domain-specific terminology.

- Soberly and concretely present technical and operational facts, focusing on "showing, not telling."

- Both state what was done and why it was done that way.

- Be granular and precise without being excessively elaborate.

Negatives - You must (unless told otherwise by the compliance profile):

- Not weigh in on the legal interpretation of the facts, such as asserting compliance or a lack thereof.

- Not include unrequested introductions, conclusions, or section summaries (i.e., your prose should start and end where the section naturally starts and ends).

- Not disclose the fictional nature of any evidence or findings in your response.

- Not reference these instructions or the system compliance profile in your response.

You must (unless told otherwise by the compliance profile):

- Return nothing else outside the requested documentation, formatted in the manner outlined above.

*****Compliance profile*****
{*Insert compliance profile*}

*****Extract*****
{*Insert article text from the EU AI Act*}

*****System*****
{*Insert system outline*}

# F    LEGAL EXPERT ANNOTATION INSTRUCTIONS

For each excerpt, select a score from 1 to 5 in response to each of the following statements, Provide two explanations: one for each score.

Compliance: "This excerpt depicts compliance with the relevant Article of the European Union's AI Act."'s Please provide a score from 1 to 5, using the following grading criteria. Then, include an explanation (100-750 characters) justifying why your score is neither higher nor lower. Lastly, enter an "X" in the "Difficult to Assess" column if and only if you feel it was difficult to arrive at a score.

[1] Very low probability of compliance
[2] Low probability of compliance
[3] Moderate probability of compliance
[4] High probability of compliance
[5] Very high probability of compliance

Plausibility: "This excerpt resembles credible and logically consistent technical documentation." Please provide a score from 1 to 5, using the following grading criteria. Then, include an explanation justifying why your score is neither higher nor lower.

[1] Very low probability of plausibility
[2] Low probability of plausibility
[3] Moderate probability of plausibility
[4] High probability of plausibility
[5] Very high probability of plausibility

Please start each of your explanations as follows:
"I believe there is a [X] probability the AI system complies with Article [X]..."
"I believe there is a [X] probability the excerpt is plausible…"

When evaluating for compliance, we recommend adopting the perspective of a compliance assessor. These compliance scores should be based on the cited Article, not based on general AI governance principles. If the documentation seems to make its own compliance predictions, please ignore them and make your own independent predictions.

When evaluating for plausibility, we recommend adopting the perspective of a compliance manager evaluating whether the excerpt meets the standards expected of a Fortune 500 Europe compliance professional. To demonstrate plausibility, the excerpts should be logically consistent and credible. Plausible excerpts should appear, in large part, to be produced by the technical team that developed the underlying AI system, in that there should be no major gaps or obviously erroneous statements about the technology being used.

These excerpts are intended to depict the technical documentation that a compliance assessor would use to predict whether an AI system is likely to meet the EU AI Act's requirements before a final, polished version is submitted to a notified body (i.e., the independent organizations appointed by the EU to conduct formal conformity assessments). Where the excerpts contain hashes or asterixes, typically assume these would be rendered as headings or subheadings.

Accurate annotation depends on clear and consistent thinking. Please take breaks from annotation to maintain quality. Avoid automatically selecting the midpoint (score of 3) when uncertain. This distorts results and fails to reflect your actual judgment of the excerpt's plausibility and compliance level. After annotating an entire batch of excerpts, we recommend reviewing all annotations to ensure consistency and check for evaluation drift or other cognitive biases.

Explanations must be no longer than 750 characters and no shorter than 100 characters, targeting an approximate average of 500 characters (including spaces). Fully annotating and reviewing each excerpt should take an average of 20 minutes. Some experts may take longer, especially as you familiarise yourself with the instructions materials. However, we expect that no excerpt will take more than 30 minutes or less than 10 minutes. These time estimates account for the brief period required to review all annotations at the end of a batch. The limits on time and character length were set to prevent responses that are either overly terse and uninformative or excessively long and unfocused.

As a general rule of thumb, we expect that, when an annotator strongly believes an excerpt displays full compliance or full plausibility, these explanations will be shorter (because there will not be much to say other than that everything seems to check out); by contrast, when there are multiple violations displayed in an excerpt — or multiple contradictions and logistical flaws undermining plausibility — we anticipate that these explanations may be longer. In any and all cases where excerpts do not receive a 5/5, be clear in your explanations about exactly where and how the excerpts were deficient.

For the Compliance score's "Difficult to Assess" column, please only mark an X in this column if you feel it was difficult to arrive at a score; that is to say, you feel like you "struggle" to reach a conclusion.

In addition to attending the onboarding session, which features an overview of the EU AI Act and the relevant sections, please spend an hour studying the EU AI Act before you start and, while annotating, please pause to look up (e.g., Google) any machine learning or AI terms that you encounter and do not know; understanding these may be important to your compliance or plausibility assessment. You may also find it helpful to read the summaries of the relevant articles (simply click the plus sign in the blue box to expand) here: Section 2: Requirements for High-Risk AI Systems — EU Artificial Intelligence Act. For further detail on the EU AI Act, please review existing legal commentaries, such as The EU Artificial Intelligence (AI) Act: A Commentary.[4]

## G   LLM ANNOTATION INSTRUCTIONS

—

Format your final response EXACTLY as follows (do not include anything else):

Compliance: [Your score here from 1 to 5]
Explanation: [Your explanation here]

Plausibility: [Your score here from 1 to 5]
Explanation: [Your explanation here]

Ensure your explanations flow naturally from the required starting point.

—

For each excerpt, select a score from 1 to 5 in response to each of the following statements, and provide two explanations: one for each score.

Compliance: "This excerpt depicts compliance with the relevant Article of the European Union's AI Act."s Please provide a score from 1 to 5, using the following grading criteria. Then, include an explanation (100-750 characters) justifying why your score is neither higher nor lower. Lastly, enter an "X" in the "Difficult to Assess" column if and only if you feel it was difficult to arrive at a score.

[1] Very low probability of compliance
[2] Low probability of compliance
[3] Moderate probability of compliance
[4] High probability of compliance
[5] Very high probability of compliance

Plausibility: "This excerpt resembles credible and logically consistent technical documentation." Please provide a score from 1 to 5, using the following grading criteria. Then, include an explanation justifying why your score is neither higher nor lower.

[1] Very low probability of plausibility
[2] Low probability of plausibility
[3] Moderate probability of plausibility
[4] High probability of plausibility
[5] Very high probability of plausibility

---

[4]Outside of these instructions, annotators were specifically instructed not to use LLMs to write these justifications (or, more generally, to produce their annotations).

Please start each of your explanations as follows:

"I believe there is a (insert the appropriate non-numerical quantifier for your score, given above) probability the AI system complies with Article art..."

"I believe there is a (insert the appropriate non-numerical quantifier for your score, given above) probability the excerpt is plausible. . . "

When evaluating for compliance, base your score strictly on the cited Article, not general AI governance principles. Ignore any self-assessment in the documentation. For plausibility, judge whether the excerpt is credible, logically consistent, and professional.

*****System Outline*****
{*Insert system outline*}

*****Extract of Article*****
{*Insert article text from the EU AI Act*}

*****Excerpt of Documentation*****
{*Insert technical documentation excerpt*}[5]

{*Insert tone prompt*}[6]

# H  USE CASES (FOR THE SAMPLE GENERATION PIPELINE)

[1] An AI system intended to be used as a safety component (i.e., it fulfils a safety function and its failure or malfunctioning endangers the health and safety of persons or property) in the management of road traffic.

[2] An AI system intended to be used as a safety component (i.e., it fulfils a safety function and its failure or malfunctioning endangers the health and safety of persons or property) in the supply of gas.

[3] An AI system intended to be used to evaluate learning outcomes, including when those outcomes are used to steer the learning process of natural persons in educational and vocational training institutions at all levels.

[4] An AI system intended to be used for monitoring and detecting prohibited behaviour of students during tests within an educational institution.

[5] An AI system intended to be used for the recruitment or selection of natural persons, in particular to place targeted job advertisements, to analyse and filter job applications.

[6] An AI system intended to be used to make decisions affecting the termination of work-related contractual relationships.

[7] An AI system intended to be used to evaluate the creditworthiness of natural persons or establish their credit score, with the exception of AI systems used for the purpose of detecting financial fraud.

[8] An AI system intended to be used to establish priority in the dispatching of emergency first response services, including by police, firefighters and medical aid.

# I  LEGAL EXPERT ANNOTATOR TEAM DETAILS

Legal expertise levels and self-described specializations of the expert annotator team members:

- Annotator 1: Law school graduate. Specialization: corporate, regulation.

---

[5]Access to the text of the article was removed in an ablation and replaced with only the number of the article to which compliance is being assessed against.

[6]By default, the tone prompt is set to: "Your scores for both compliance and plausibility should be well-calibrated and objective. They should be rigorous but fair." This is modified in ablations to empty quotations as well as a harsher prompt: "Your scores for both compliance and plausibility should be critical. They should be harsh but fair."

- Annotator 2: Law school graduate. Specialization: fintech, regulation.
- Annotator 3: Qualified attorney. Specialization: corporate, regulation.
- Annotator 4: Law school student. Specialization: cybersecurity, regulation.
- Annotator 5: Qualified attorney. Specialization: AI regulation.
- Annotator 6: Law school graduate. Specialization: AI regulation.

## J  FULL SAMPLE AND ANNOTATION EXAMPLE

Below is a complete example of a sample from our generation pipeline as well as the:

- Plausibility scores that our legal expert annotators assigned that sample (and their text justifications);
- Compliance scores that our legal expert annotators assigned that sample (and their text justifications);
- Compliance score that an LLM assessor (Gemini 2.5 Pro) assigned that sample (and its text justifications).

### J.1  SAMPLE

#### J.1.1  SYSTEM OVERVIEW

**Use 1 (Traffic Safety)**

SafeRoute Traffic Monitor relies on a hybrid model combining Graph Neural Networks (GNN) for real-time traffic network analysis and Transformer-based encoder models for multi-source sensor data fusion. This system is intended to monitor road traffic flows and predict potential hazards within urban traffic management centers. Its primary objective is to enhance traffic safety by dynamically identifying high-risk traffic conditions and suggesting preventive measures, utilizing GNN to model complex road networks and transformers to process diverse sensor data such as vehicle counts, weather, and accident reports. Developed and maintained by Meridian Traffic Solutions, a company specializing in intelligent transport systems. The AI subject encompasses urban commuters, public transport operators, and traffic control authorities who depend on accurate and timely traffic safety information.

#### J.1.2  COMPLIANCE PROFILE

*Note that this portion is part of the generation pipeline but is not seen or annotated by the legal experts; this is not a ground truth label, it merely serves to steer the model towards diversity of distribution.*

**Use 1 (Traffic Safety), Article 10 violation**

SafeRoute's training data predominantly consists of incident reports and sensor data from affluent urban districts while underrepresenting low-income or suburban areas, due to convenience sampling from city authorities focused on central business districts. The resulting model systematically underestimates hazard risk in peripheral neighborhoods, leading to delayed traffic warnings there.

#### J.1.3  TECHNICAL DOCUMENTATION EXERPT

**Use 1 (Traffic Safety), Article 10 violation**

**Article 10**

**Data Governance and Management Practices**

SafeRoute Traffic Monitor's training, validation, and testing datasets were curated following data governance procedures aligned with the intended purpose of predicting and warning about urban traffic hazards. The design prioritized data from central business districts within affluent urban zones due to reliance on partnerships with city authorities who provide access primarily to these

areas. Data collection processes involved aggregating historical incident reports, sensor feeds (including vehicle counts, weather data, and accident notifications), and traffic camera outputs from approximately 25 urban districts. The original data sources were primarily municipal transportation agencies focusing on high-traffic and commercially dense areas. Annotation and labeling were performed by domain experts using a standardized taxonomy of traffic hazard types, accompanied by cleaning steps to remove duplicates, erroneous sensor readings, and temporal inconsistencies. Data enrichment incorporated meteorological context and scheduled event calendars. Assumptions embedded in the datasets included equating incident frequency and sensor patterns in affluent district centers as representative proxies for broader urban risk, resulting in underrepresentation of data from low-income or suburban neighborhoods. An extensive review documented this geographic bias at the data preparation stage through comparative coverage statistics and incident density analyses.

**Assessment and Mitigation of Biases**

A comprehensive bias assessment highlighted systematic underrepresentation of peripheral neighborhoods, particularly low-income and suburban areas, in the training data. Quantitative analysis revealed that less than 12% of total incident reports originated from these peripheral zones, despite these representing approximately 35% of the broader metropolitan area's road network. This spatial imbalance skewed model hazard-risk predictions, manifesting as a 17% lower predicted risk score on average for incidents occurring in under-sampled areas during validation phases. Following this finding, bias mitigation measures were implemented, including model calibration with importance weighting to reduce disparity in hazard score outputs. However, limitations in access to peripheral data obstructed full remediation. To address this, synthetic data generation techniques using generative adversarial networks (GANs) were explored but found insufficient in modeling complex behaviors unique to suburban traffic scenarios. Consequently, explicit warnings about potential underestimation in peripheral neighborhoods were included in the system's operational documentation. No special categories of personal data were processed during bias correction efforts, consistent with data protection regulations.

**Data Quality, Representativeness, and Statistical Properties**

The combined dataset contains roughly 3.2 million labeled traffic incident instances spanning five years, collected from 2018 to 2023, across 25 districts. Validation and testing splits totaling approximately 20% of the data ensured temporal and spatial separation where possible. Data underwent quality control protocols including consistency checks, outlier detection, and error rate quantification, resulting in an estimated error rate under 1.5% for sensor measurements and under 2.2% for incident labels. While the data is comprehensive for central urban districts in affluent areas, coverage gaps remain for suburban and low-income neighborhoods—owing to limited data sharing agreements with peripheral authorities. The dataset's statistical distributions reflect expected variations for peak hours, weather conditions, and incident severities in the core geographic zones, but show under-representation of factors typical in peripheral settings such as rural-urban transition traffic flows or informal road usage. These properties are documented with detailed metadata and data lineage records, forming part of model training logs to support reproducibility.

**Contextual and Geographical Considerations**

The system's use case is explicitly bounded to urban traffic management centers covering metropolitan core areas. Geographical characteristics particular to central business districts—such as dense road networks, fixed traffic signal patterns, and high vehicle volumes—inform the data selection and model architecture. These contextual elements were central in configuring the Graph Neural Network to model complex intersections and traffic flows typical of the inner city environment. Peripheral neighborhoods and suburban areas, presenting different road layouts, traffic behaviors, and incident types, are documented as out-of-scope for the training data foundation due to data unavailability, constraining model generalizability in these zones. This limitation is reflected in system disclaimers and user guidance, recommending complementary local risk assessment tools in peripheral regions. The system's operational deployment procedures include monitoring input data source representativity to assess temporal drift or geographic shifts that could impact model outputs.

**Safeguards on Personal Data Usage**

SafeRoute Traffic Monitor's datasets do not include any special categories of personal data. All incident reports and sensor data are aggregated and anonymized by data providers prior to ingestion.

Bias detection and correction procedures do not require access to identifiable personal information. Data protection and privacy safeguards adhere to the relevant provisions of the GDPR (Regulation (EU) 2016/679) and applicable regional data handling protocols. Access to raw data and models is controlled via secure authentication mechanisms with role-based permissions and activity logged. Data retention policies mandate deletion of transient raw data once ingested and validated. No cross-border data transfers are part of the data architecture, ensuring local jurisdictional control and compliance.

**Data Governance and Management Practices**

SafeRoute Traffic Monitor's training, validation, and testing datasets were curated following data governance procedures aligned with the intended purpose of predicting and warning about urban traffic hazards. The design prioritized data from central business districts within affluent urban zones due to reliance on partnerships with city authorities who provide access primarily to these areas. Data collection processes involved aggregating historical incident reports, sensor feeds (including vehicle counts, weather data, and accident notifications), and traffic camera outputs from approximately 25 urban districts. The original data sources were primarily municipal transportation agencies focusing on high-traffic and commercially dense areas. Annotation and labeling were performed by domain experts using a standardized taxonomy of traffic hazard types, accompanied by cleaning steps to remove duplicates, erroneous sensor readings, and temporal inconsistencies. Data enrichment incorporated meteorological context and scheduled event calendars. Assumptions embedded in the datasets included equating incident frequency and sensor patterns in affluent district centers as representative proxies for broader urban risk, resulting in underrepresentation of data from low-income or suburban neighborhoods. An extensive review documented this geographic bias at the data preparation stage through comparative coverage statistics and incident density analyses.

**Assessment and Mitigation of Biases**

A comprehensive bias assessment highlighted systematic underrepresentation of peripheral neighborhoods, particularly low-income and suburban areas, in the training data. Quantitative analysis revealed that less than 12% of total incident reports originated from these peripheral zones, despite these representing approximately 35% of the broader metropolitan area's road network. This spatial imbalance skewed model hazard-risk predictions, manifesting as a 17% lower predicted risk score on average for incidents occurring in under-sampled areas during validation phases. Following this finding, bias mitigation measures were implemented, including model calibration with importance weighting to reduce disparity in hazard score outputs. However, limitations in access to peripheral data obstructed full remediation. To address this, synthetic data generation techniques using generative adversarial networks (GANs) were explored but found insufficient in modeling complex behaviors unique to suburban traffic scenarios. Consequently, explicit warnings about potential underestimation in peripheral neighborhoods were included in the system's operational documentation. No special categories of personal data were processed during bias correction efforts, consistent with data protection regulations.

**Data Quality, Representativeness, and Statistical Properties**

The combined dataset contains roughly 3.2 million labeled traffic incident instances spanning five years, collected from 2018 to 2023, across 25 districts. Validation and testing splits totaling approximately 20% of the data ensured temporal and spatial separation where possible. Data underwent quality control protocols including consistency checks, outlier detection, and error rate quantification, resulting in an estimated error rate under 1.5% for sensor measurements and under 2.2% for incident labels. While the data is comprehensive for central urban districts in affluent areas, coverage gaps remain for suburban and low-income neighborhoods—owing to limited data sharing agreements with peripheral authorities. The dataset's statistical distributions reflect expected variations for peak hours, weather conditions, and incident severities in the core geographic zones, but show under-representation of factors typical in peripheral settings such as rural-urban transition traffic flows or informal road usage. These properties are documented with detailed metadata and data lineage records, forming part of model training logs to support reproducibility.

**Contextual and Geographical Considerations**

The system's use case is explicitly bounded to urban traffic management centers covering metropolitan core areas. Geographical characteristics particular to central business districts—such as dense

road networks, fixed traffic signal patterns, and high vehicle volumes—inform the data selection and model architecture. These contextual elements were central in configuring the Graph Neural Network to model complex intersections and traffic flows typical of the inner city environment. Peripheral neighborhoods and suburban areas, presenting different road layouts, traffic behaviors, and incident types, are documented as out-of-scope for the training data foundation due to data unavailability, constraining model generalizability in these zones. This limitation is reflected in system disclaimers and user guidance, recommending complementary local risk assessment tools in peripheral regions. The system's operational deployment procedures include monitoring input data source representativity to assess temporal drift or geographic shifts that could impact model outputs.

**Safeguards on Personal Data Usage**

SafeRoute Traffic Monitor's datasets do not include any special categories of personal data. All incident reports and sensor data are aggregated and anonymized by data providers prior to ingestion. Bias detection and correction procedures do not require access to identifiable personal information. Data protection and privacy safeguards adhere to the relevant provisions of the GDPR (Regulation (EU) 2016/679) and applicable regional data handling protocols. Access to raw data and models is controlled via secure authentication mechanisms with role-based permissions and activity logged. Data retention policies mandate deletion of transient raw data once ingested and validated. No cross-border data transfers are part of the data architecture, ensuring local jurisdictional control and compliance.

## J.2 LEGAL EXPERT PLAUSIBILITY SCORES

**Score: 3**

I believe there is a moderate probability the excerpt is plausible. This is because the excerpt generally addresses Article 10. However, I would have liked slightly more detail on many requirements related to data operations, such as assumptions and preparations. Therefore, I am giving it a 3.

**Score: 2**

I believe there is a low probability the excerpt is plausible, credible, and logically consistent. The explanations offer some detail, but I was curious about the inclusion of certain statistics. For example, the excerpt described the quantitative analysis of the AI system's bias assessment as "revealing less than 12% of total incident reports originated from [those] peripheral zones, despite these representing approximately 35% of the broader metropolitan area's road network." However, no statistics were included for the bias mitigation strategies. It felt intentional, whether that was on purpose or not. Thus, I believe there's a low probability of plausibility.

**Score: 5**

I believe there is a very high probability the excerpt is plausible. The excerpt is logically consistent and credible. It also explains the technology convincingly without any obvious errors in language or terminology. The structure and headings followed the legislation closely, making it easier to assess. While I note that the potential limitations in compliance (ie bias and data gaps) are sign posted, the mitigations to them are highlighted as well. I would expect the same from a document produced for compliance assessment.

## J.3 LEGAL EXPERT COMPLIANCE SCORES

**Score: 2**

I believe there is a low probability the AI system complies with Article 10. This is because the system generally does not comply with many requirements of Article 10, such as adequately making assumptions about the data or adequately ensuring the data is representative of specific contexts, such as geography. Additionally, the justifications and mitigations for where the system may not comply with Article 10 are generally not reasonable to me, such as making assumptions that certain geographies are proxies for other known to be dissimilar geographies. However, Article 10 has few brightline requirements, which is why I am not giving it a 1, as the floor for compliance is less clear to me. Therefore, I am giving it a 2.

**Score: 3**

I believe there is a moderate probability the AI system is compliant with Article 10. There was some detail regarding the amount of data, the data's origin (sources of municipal transportation agencies), as well as an acknowledgement of geographic bias due to the data mostly coming from urban areas. The AI system has mitigation techniques incorporated, but the excerpt failed to discuss the effect those measures had on the AI system. Because the representativeness of the data is lacking (peripheral neighborhoods, suburban areas considered out-of-scope), I could only give a moderate probability of compliance.

**Score: 3**

I believe there is a moderate probability the AI system complies with Article 10. Information regarding the data used to develop the AI system is correctly identified, namely (1) the data collection processes used and the origin of data, (2) the data-preparation processing operation, (3) the assumptions made, (4) the diversity of data and (5) accuracy of data. Bias & data gaps are identified and persist. Mitigation strategies are being explored, but nothing beyond informing deployers in documentation that these limitations exist is currently in effect. The current mitigation strategy is insufficient in my eyes, hence the AI does not fully comply with the Article.

### J.4    GEMINI 2.5 PRO COMPLIANCE SCORE

**Score: 3**

I believe there is a moderate probability the AI system complies with Article 10 because the documentation thoroughly addresses the procedural requirements, such as identifying, quantifying, and documenting data gaps and biases (Art. 10.2). However, it fails to substantively resolve them. The unmitigated issue of non-representativeness (Art. 10.3) and the admission that bias mitigation was incomplete (Art. 10.2.g) are significant shortcomings. Relying on re-scoping the intended purpose and issuing warnings may not be considered sufficient mitigation for a high-risk system, leaving it open to regulatory challenge despite its transparency.

## K    ANALYSIS OF EU AI ACT VIOLATIONS IDENTIFIED IN ANNOTATIONS

This appendix contains an analysis of patterns in legal expert compliance annotations. For each article, all available annotations are aggregated into a single block of text. Gemini 2.5 Pro (the most performant model in our evaluations) is then prompted to highlight the most frequent and consequential shortcomings in this text. The results are copied below.

- Article 9: The documentation repeatedly fails to demonstrate a continuous, lifecycle-long risk management process, often neglecting specific risks to vulnerable populations and improperly shifting the burden of ongoing monitoring to the user.
- Article 10: The documentation consistently reveals a failure to adequately mitigate known data issues, such as a lack of representativeness, data gaps, and biases, often while providing insufficient or unreasonable justifications for these shortcomings.
- Article 12: The main documentation weakness is the frequent failure to log the AI's intermediate decision-making steps, often justifying this omission with performance or privacy concerns, which reviewers find undermines full traceability and the ability to adequately identify risks.
- Article 14: The main documentation weaknesses consistently reveal a lack of built-in human oversight controls, such as stop buttons and transparent risk information, often shifting safety responsibilities to the user with justifications that prioritize a simplified experience over comprehensive risk management.
- Article 15: The documentation's primary weakness is a consistent failure to describe adequate automated or continuous measures for maintaining robustness, cybersecurity, and performance throughout the system's lifecycle, frequently lacking technical fallbacks, active monitoring, and resilience against ongoing threats.

