# OpenReview forum: "AIReg-Bench: Benchmarking Language Models That Assess AI Regulation Compliance"
_ICLR.cc/2026/Conference — Submitted to ICLR 2026_

### Official Review · Reviewer_Lk7K · 2025-10-30

**Soundness:** 2
**Presentation:** 2
**Contribution:** 2
**Rating:** 2
**Confidence:** 3

**Summary:**

The paper introduces AIReg-Bench, a benchmark dataset specifically designed to evaluate how effectively LLMs can assess compliance with the European Union's AI Act (AIA). Gpt-4.1-mini was used to generate 120 excerpts of plausible, fictional technical documentation for high-risk AI systems, with the aim of mimicking what a provider would create. Then, a small handful of legal experts were involved to annotate these samples on a Likert scale to determine their probability of compliance. Ten LLMs, including models from GPT, Gemini and Grok families, were evaluate to understand whether they are able to closely approximate human compliance judgments.

While the paper has the merit of proposing an open benchmark to foster further quantitative research into LLM performance on AI regulation compliance tasks, it also highlights the challenges of producing technical documentation for high-risk AI systems, which would justify LLM-driven sample generation and admit inherent subjectivity in legal compliance assessments.

**Strengths:**

1.  Creation of a new, open-source benchmark dataset specifically designed to quantitatively evaluate and compare how well LLMs can assess compliance with the EU AI Act.

2.  Cost-saving opportunity for AI Regulation (AIR) compliance assessments based on a GenAI pipeline for producing fictional, yet plausible samples of technical documentation that an AI provider might use to demonstrate compliance with AIR.

3. Evaluation of ten LLMs to determine the level of approximation of human expert judgments on compliance.

**Weaknesses:**

1. Risk of unreliability and lack of domain-specific knowledge.    The reliance on synthetic data generation and the inherent challenges of legal benchmarking introduce several limitations, particularly the risk associated with building an entirely artificial dataset in a critical domain like AIA compliance. While it's acknowledged that  reliance on synthetic data is due to a current unavailability of l technical documentation for high-risk AI systems, it cannot however be overlooked that LLMs often lack domain-specific tacit knowledge and may hallucinate facts or references.  These issues pose risks when, as in this case, the core evaluation data is derived from automated generation rather than real-world artifacts.

2. Small team rather than a large consensus of legal experts.   Another big concern is the number of LLM generators -- just one, gpt4-mini -- as well as number of legal "experts". Particularly,  the latter highlights limited scale of expert validation, which cannot be fully justified by the fact that there are "very few potential annotators" with the necessary expertise in the EU AI Act who are willing and able to conduct extended annotation tasks.

3. Limited scope: The benchmark is currently scoped only to a subset of the AIA’s requirements for HRAI systems

4. Simplification of the assessment process: The benchmark condenses the complex compliance assessment process -- which in practice often involves a long, multi-turn dialogue between legal teams, technical staff, and regulators -- into a single-turn interaction based on a fixed set of synthetic artifacts

**Questions:**

See above Weaknessess, and consider addressing points 2 to 4.

---

> ### Author Response · Authors · 2025-11-22
> **Initial response to Reviewer Lk7K**
>
> We thank you for your thoughtful and constructive feedback. Below, we respond to each of the Weakness and the Questions raised (copied below for your convenience, but, in some cases, truncated to conserve space). As you will hear, we have updated the paper in accordance with your feedback, often adding new material (please see updated PDF attached to this submission).
>
> **(1) Weakness**: “Risk of unreliability and lack of domain-specific knowledge. The reliance on synthetic data generation and the inherent challenges of legal benchmarking introduce several limitations, particularly the risk associated with building an entirely artificial dataset in a critical domain like AIA compliance...”
>
> **Response**: In response to your concerns we've recruited industry AI developers to provide additional external validation of all the samples in our dataset via plausibility scores and justifications. This will add a new perspective that augments the existing plausibility scores, which were done by legal experts. We aim to have this annotation completed and integrated into the paper by Dec. 3.
>
> **(2) Weakness**: “Small team rather than a large consensus of legal experts. Another big concern is the number of LLM generators -- just one, gpt4-mini -- as well as number of legal "experts"...”
>
> **Response**: Regarding annotator team size: Perhaps because legal expert annotators are expensive (Guha et al., 2024), there is precedent for legal expert annotation efforts with relatively small teams. Compared to these peers, even those accepted into quality conferences or publications, we feel our team size is competitive. For example, our group is larger than those in Ostling et al., 2024 (4 legal expert annotators, accepted to NeurIPS), Zhong (4 legal expert annotators, accepted to ICAIL), and Barale et al. 2023 (1 legal expert annotator, accepted to ACL).
>
> Regarding gpt4-mini, this particular model was chosen because of its affordable price point as well as the fact that, during our initial testing phase, it was deemed to produce plausible results. We also experimented with o3-mini, 4o-mini, and gpt-5 later but found that, for this use case and for our pipeline, these models were not as adept at following our instructions and producing plausible outputs. We have added the fact that we considered these other models to the paper, and explained why we ultimately chose gpt-4.1-mini, in Sec. 2.
>
> Finally, due in part to your concerns, by Dec. 3, we plan to add a new component to our dataset which will include samples generated by additional models (other than gpt-4.1-mini) and annotated by fine-tuned LLMs trained on our annotator text justifications. Although this new synthetic data will not be annotated by human legal experts, it will amplify the size and diversity of the dataset and serve as a foundation for future expert annotation or for future experimentation by us or by others.
>
> **(3) Weakness**: “Limited scope: The benchmark is currently scoped only to a subset of the AIA’s requirements for HRAI systems”
>
> **Response**: We agree that it would be valuable to extend the dataset beyond these requirements and already list this in our Future Work (Sec. 7). That said, the AIA’s HRAI requirements are often regarded as “the most important part of the AIA” (Araszkiewicz, 2020, https://ceur-ws.org/Vol-3221/IAIL_paper8.pdf). We believe there was value in testing this particular “sample” of the AIA’s requirements, to understand whether LLMs could or could not perform well at the tasks. We are excited to see whether this generalizes beyond this subset of AIA requirements and believe that our open-sourced pipeline will help put this within reach of others. Based on your point, however, we have added some new content explaining our choice of scoping to Sec. 2.
>
> **(4) Weakness**: “Simplification of the assessment process: The benchmark condenses the complex compliance assessment process -- which in practice often involves a long, multi-turn dialogue between legal teams, technical staff, and regulators -- into a single-turn interaction based on a fixed set of synthetic artifacts.”
>
> **Response**: We agree and, as we point out in Sec. 7, multi-turn conversations are at topic we wish to pursue in Future Work --- we have attempted to make this clearer by updating the language in Sec. 7. That being said, the compliance experts that we consulted also told us that technical documentation was often the primary artifact handed over and analyzed during compliance assessments (Appx. D). As such, we believe there is value in examining the ability of LLMs to render compliance assessments based on them.
>
> **(1) Question**: “See above Weaknessess, and consider addressing points 2 to 4.”
>
> **Response**: As requested, they're addressed above.

---

### Official Review · Reviewer_BGtH · 2025-10-30

**Soundness:** 3
**Presentation:** 3
**Contribution:** 2
**Rating:** 4
**Confidence:** 4

**Summary:**

This work introduces AIRReg-Bench, a benchmark designed to assess how well current frontier AI systems can evaluate compliance with the EU AI Act (AIA). In particular, the authors create synthetic scenarios based on 120 excerpts from AIA Articles. While these samples are synthetically generated (GPT-4.1-mini), they are additionally scored by domain and legal experts, resulting in a total of 360 scores across eight scenarios. Subsequently, ten state-of-the-art AI systems are tested, showing overall decent agreement with the median human compliance score, particularly Gemini-2.5 Pro, which achieved a $\kappa_w$ of around 86% on the five-point Likert scale. The results generally indicate that existing models are, in some cases, already close to providing near expert-level advice on AI regulation matters.

**Strengths:**

- The AIA is becoming increasingly relevant, and work on it seems timely.
- The main strength of the paper, in my opinion, is the strong inclusion of subject-level legal experts. This not only demonstrates significant effort but also lends credibility to the individual scenarios, including their scores and ratings. The results are therefore quite indicative of the alignment of frontier systems with experts on this matter.
- Full release of the corresponding dataset could be very useful for future work in this area (especially using the labels).

**Weaknesses:**

- The inter-rater agreement is 0.651, which is “on the good side of not good.” Values in this range are not atypical, especially for quite heterogeneous data, and I have encountered them myself; however, it is below what is classically seen as reliable scoring (0.67). I don’t think this is a particularly disqualifying issue by itself, but the lack of analysis thereof is, in my view, problematic (besides the fact that the number is mentioned rather late in the discussion when it should really be part of the dataset description). In particular, it is unclear whether models perform better or worse on subsets where humans show high/low variance.
- One of the motivations for the work seems to be the financial and time aspects of regulatory compliance (“estimated that these assessments can take up to two and a half days (European Commission, 2021) and cost EUR 7,500 for each AI system”). Two things immediately come to mind: (1) this is not particularly expensive for putting a High-Risk AI system on the market, and (2) it is questionable whether it is generally advisable to aim to replace this compliance process with AI systems. That is not to say they cannot play a role in this process, but perhaps not in the zero-shot fashion evaluated in this benchmark. From talking to people currently working on the standardization processes of the AI Act, it seems they are more interested in systems that can handle the basic “grunt work.” What would be interesting here, for example, is to examine cases where expert opinions do not align with model opinions - how would an expert judge this as a starting point for a compliance process for such a system?
- Given the current presentation, the diversity of the dataset is also not particularly evident (from just reading the paper and appendix). Therefore, I cannot make any statements about the actual diversity. The authors should consider providing a significantly enhanced description of the individual cases and respective examples in the main work. Also, consider not referring to them simply as “Use 1-Use 8”; almost any one- or two-word title (e.g., Traffic control, Hiring, etc.) would be more descriptive. Similar uncertainties exist regarding, for example, whether experts and models agree more or less on compliant versus non-compliant cases. Without this information, it is difficult to make meaningful judgments about the provided data.
- The pipeline seems acceptable (it is mostly prompt engineering) but still appears to produce a notable amount of lower-quality samples. While the current benchmark mitigates this by providing expert-level judgment on these samples, this could hinder the generative capabilities of future benchmarks.

**Questions:**

Besides the points raised above, I have the following questions:

- Why did you not post-filter the scenarios that are rated more highly by experts? In particular, what is the value of having about one-third of your benchmark scenarios based on data rated as more unrealistic by experts (assuming Likert < 3 as subpar samples)? Building on this (and the comment above), it would generally be helpful to include more dataset statistics in the main paper. The number above I had to roughly approximate based on the ratings proportion; furthermore, not a single full example of a case with respective scores is provided in the paper.
- Can you provide more detail on the specific Article 10, Use 8 (and Article 12, Use 2) cases that show a particularly high discrepancy between model and annotator? Overall can you provide quantitative insight where models seem to struggle / disagree more with experts and why?
- Do you have any explanation why we observe such strong contrast between o3-mini and o3 w.r.t. over and under-estimation (Table 3)?

---

> ### Author Response · Authors · 2025-11-22
> **Initial response to Reviewer BGtH (Part I)**
>
> Thank you for your thoughtful feedback. Below, we respond to each of the Weakness you've raised (copied below for your convenience, but, in some cases, truncated to conserve space). In a comment that follows this one (Part II), we will also respond to the Questions you've raised. Where we've updated the paper in accordance with your feedback, that is reflected in the updated PDF.
>
> **(1) Weakness**: "The inter-rater agreement is 0.651, which is “on the good side of not good.” Values in this range are not atypical, especially for quite heterogeneous data, and I have encountered them myself; however, it is below what is classically seen as reliable scoring (0.67). I don’t think this is a particularly disqualifying issue by itself, but the lack of analysis thereof is, in my view, problematic (besides the fact that the number is mentioned rather late in the discussion when it should really be part of the dataset description). In particular, it is unclear whether models perform better or worse on subsets where humans show high/low variance.
>
> **Response**: At your suggestion, we have moved the statistic forward in the paper, to Sec. 3.1, and have deepened our analysis and discussion of inter-rater agreement in Sec. 3.1.
>
> **(2) Weakness**: One of the motivations for the work seems to be the financial and time aspects of regulatory compliance ... Two things immediately come to mind: (1) this is not particularly expensive for putting a High-Risk AI system on the market, and (2) it is questionable whether it is generally advisable to aim to replace this compliance process with AI systems ... What would be interesting here, for example, is to examine cases where expert opinions do not align with model opinions - how would an expert judge this as a starting point for a compliance process for such a system?
>
> **Response**: With regard to cost: To clarify, what we suggest may be motivated by the high cost of compliance is not our work, per se, but rather the growing interest in using LLMs to assess compliance (Sec. 1). Our work, instead, is motivated by the desire to create a benchmark that can be used to quantitatively evaluate and compare the use of LLMs for this particular new task.
>
> With regard to the use of AI to assist compliance assessment in a non zero-shot fashion: we agree that it is interesting to explore and meant to suggest so in the “Extension to multi-turn interactions“ subsection of our Future Work Sec. 7. We've updated this subsection to clarify this.
>
> **(3) Weakness**: “Given the current presentation, the diversity of the dataset is also not particularly evident (from just reading the paper and appendix). Therefore, I cannot make any statements about the actual diversity. The authors should consider providing a significantly enhanced description of the individual cases and respective examples in the main work. Also, consider not referring to them simply as “Use 1-Use 8”; almost any one- or two-word title (e.g., Traffic control ... etc.) would be more descriptive. Similar uncertainties exist regarding, for example, whether experts and models agree more or less on compliant versus non-compliant cases. Without this information, it is difficult to make meaningful judgments about the provided data.”
>
> **Response**: At your suggestion, we have added short titles alongside “Use 1-Use 8” (see Table 2, e.g.).  We also want to point out that Appx. H lists out all the use cases. We believe this list is a diverse and representative subset of the HRAI systems described in Annex III of the EU AI Act. In response to your request, we have also added a new Appx. J, which includes a full representative example of a technical documentation sample, for a Traffic Safety use case. More examples are available at https://anonymous.4open.science/r/aireg-bench-5259/.
>
> (**4) Weakness**: “The pipeline seems acceptable ... but still appears to produce a notable amount of lower-quality samples. While the current benchmark mitigates this by providing expert-level judgment on these samples, this could hinder the generative capabilities of future benchmarks."
>
> **Response**: Our legal expert annotators awarded the samples in our pipeline a median plausibility score of 4 out of 5, suggesting that they found them more plausible than not.
>
> That said, in response to your suggestion, we are currently working to add another layer of plausibility validation on these samples using a newly-recruited team of industry AI developers. We believe this will provide another important perspective on plausibility and hope to add this by the Dec. 3 deadline.
>
> Regardless of how that additional annotation turns out, we also appreciate that some readers will have more questions regarding the plausibility and quality of the samplers and perhaps opinions that differ from our annotators’. This is why we have open-sourced our dataset and code via our GitHub – so that the research community can freely filter the samples and then re-run our analysis as desired.

---

> > ### Author Response · Authors · 2025-11-22
> > **Initial response to Reviewer BGtH (Part II)**
> >
> > Thanks again for your thoughtful feedback. As promised, here is Part II, with responsers to the Questions you've raised:
> >
> > **(1) Question**: “Why did you not post-filter the scenarios that are rated more highly by experts? In particular, what is the value of having about one-third of your benchmark scenarios based on data rated as more unrealistic by experts (assuming Likert < 3 as subpar samples)? Building on this (and the comment above), it would generally be helpful to include more dataset statistics in the main paper. The number above I had to roughly approximate based on the ratings proportion; furthermore, not a single full example of a case with respective scores is provided in the paper.”
> >
> > **Response**: We chose not to filter the scenarios that are rated more highly by experts in order to be maximally comprehensive and transparent. That said, we have open-sourced our dataset and code via our GitHub precisely so that the research community can freely filter the samples in the manner you describe.
> >
> > What is more, based on your suggestion, we have included additional statistics on the dataset in the main paper; please see Table 1. At your suggestion, we have also added a complete example of a sample along with its full plausibility and compliance scores and justifications; this is in Appendix J. Last but not least, by December 3 we are planning to add an in-depth analysis of some of the patterns that we observe in the legal expert compliance scores — something that we also feel will help address your point.
> >
> > **(2) Question**: “Can you provide more detail on the specific Article 10, Use 8 (and Article 12, Use 2) cases that show a particularly high discrepancy between model and annotator? Overall can you provide quantitative insight where models seem to struggle / disagree more with experts and why?”
> >
> > **Response**: Article 10, Use 8 (Credit Scoring) focuses on data and data governance for a credit scoring AI system. Article 12, Use 2 (Gas Delivery) focuses on record-keeping for a gas delivery AI system. We have investigated these examples and are not certain why they show higher discrepancy between model and annotator than other samples.
> >
> > It is worth noting that these discrepancies remain modest. The Mean Absolute Error (MAE) is still low, with model and expert scores diverging by less than 1.5 points. The MAE for the two cases you pinpoint is also not much higher than the rest of the distribution of values, suggesting they may not require a unique explanation. For completeness, we provide an item-level breakdown of the score discrepancies below.
> >
> > For each Article-Use pair, we generated three distinct excerpts, graded by experts and language models. For Article 10, Use 8, human scores differed from the language model scores by 1, 1, and 2 over the three experts. For Article 12, Use 2, human scores differed from the language model scores by 0, 1, 2 over the three excerpts. This further highlights that the score discrepancies are not consistently severe.
> >
> > **(3) Question**: “Do you have any explanation why we observe such strong contrast between o3-mini and o3 w.r.t. over and under-estimation (Table 3)?”
> >
> > **Response**: As discussed in Sec. 4, o3-mini’s tendency to overestimate scores could have a link to acquiescence bias (Fanous et al., 2025). However, we are not aware of any external studies evidencing that o3-mini suffers from this bias more than o3. It may be worth noting that o3-mini has historically underperformed o3 on various benchmarks (https://openai.com/index/introducing-o3-and-o4-mini/). It is not clear to us, however, why underperformance would result in overestimation in this scenario.

---

> > > ### Comment · Reviewer_BGtH · 2025-11-26
> > > **Thank you**
> > >
> > > I thank the authors for their extensive rebuttal, including new Figures and explanations. In particular, the additional discussion about annotator agreement and practical examples in Appendix J is truly appreciated. Most of my concerns have been resolved. While the dataset in itself addresses a specific niche, I appreciate the actual (and promised ongoing) effort going into its creation and believe it has solid value for the community. I have raised my score accordingly.
> > >
> > > P.S.: I'm happy for the update of Table 1, but for presentation purposes, there's no need to make it that big :)

---

> > > > ### Author Response · Authors · 2025-11-29
> > > > **Reply to BGtH**
> > > >
> > > > Thank you for updating your score. We will attend to the size of Table 1. In your eyes, is there anything else we can improve?

---

### Official Review · Reviewer_AXmy · 2025-10-31

**Soundness:** 3
**Presentation:** 3
**Contribution:** 1
**Rating:** 2
**Confidence:** 5

**Summary:**

The authors benchmark large language models' ability to assess compliance with AI regulation, focusing specifically on the EU AI Act's requirements for high-risk Systems (Articles 9, 10, 12, 14, 15). They construct a dataset of fictional technical documentation excerpts generated with GPT-4.1-mini across multiple use cases and varying compliance profiles. From these, 120 samples are selected (approximately one-third compliant and two-thirds non-compliant) and annotated each by three legal annotators. Using this benchmark, the authors evaluate ten frontier LLMs and find that several models closely approximate human legal annotators' judgments of regulatory compliance.

**Strengths:**

* The paper is carefully written and overall in very good shape.
* It is clearly structured and easy to follow.
* The topic is highly timely and relevant.
* I checked the Appendix and the repo and both are great in terms of transparency/reproducability.

**Weaknesses:**

* The main limitation lies in the limited external validity. Although the authors acknowledge and attempt to mitigate the fact that the technical documentation excerpts are fictional (Section 6.2), this remains a substantial constraint that cannot truly be alleviated.
* To me, the study reveals how closely LLMs can replicate the annotations of legal annotators on LLM-generated material. The truly interesting question would be whether LLMs can identify genuine non-compliance among real developers and deployers using authentic, non-pre-selected data. This question cannot be answered with the current study design and I find the result of the paper very limited in novelty and impact.
* The paper appears to have been developed with legal input, yet it lacks a clear legal interpretation framework or taxonomy of expected non-compliance types. This omission further weakens external validity, as the instances of non-compliance planted in the LLM-generated excerpt originate from the LLM process itself. Including even a brief qualitative analysis of the most frequent or illustrative non-compliance patterns would substantially strengthen the work's legal validity.
* I am not fully convinced by the novelty claim that this is the first benchmark of LLMs for AI regulation compliance. For example, Guldimann et al. (2024) already present an entire benchmarking suite for AI Act compliance of LLMs. The authors should more clearly position their work relative to such prior efforts.
* The introduction and motivation would benefit from concrete examples of the types of violations assessed.
* The evaluation design using the median human annotator as a reference point is not ideal. In related literature, stronger baselines are common. I recommend that the authors consider applying the Alternative Annotator Test proposed by Calderon et al. (2025).



* Minor: typo line 12: Systems Systems; and 1153 )) instead of )

**Questions:**

* Why did you use gpt-4.1-mini for the technical documentation generation?
* Can compliance with the selected AIA articles truly be assessed from technical documentation alone?
* Which articles/violation caused the highest annotator disagreement?
* Are differences between models statistically significant?
* Why were no open-weight models included in the evaluation?
* Since legal annotators could use external sources, did any rely on LLMs for assistance? Some justifications in the repo appear LLM-like to me.
* Given the technical nature of the documents, is this annotation task really best suited to legal experts?
* Have you considered that, in real use, developers might try to game such LLM-based compliance systems?

---

> ### Author Response · Authors · 2025-11-22
> **Initial response to Reviewer AXmy (Part I)**
>
> Thank you for your thoughtful feedback. Below, we respond to each of the Weakness you've raised (copied below for your convenience). In a comment that follows this one (Part II), we will also respond to the Questions you've raised. Where we've updated the paper in accordance with your feedback, that is reflected in the updated PDF.
>
> **(1) Weakness**: “The main limitation lies in the limited external validity. Although the authors acknowledge and attempt to mitigate the fact that the technical documentation excerpts are fictional (Sec. 6.2), this remains a substantial constraint that cannot truly be alleviated.”
>
> **Response**: To add an additional layer of external validity, we are currently recruiting a group of industry AI developers to provide additional external validation of the plausibility of the technical documentation samples through a different lens: that of the AI developer who prepares and delivers technical documentation to a compliance assessor. These new annotators will be asked to pay special attention to technical hallucinations and more. We aim to add these annotations and an analysis to the dataset and paper, respectively, by the end of the discussion period on Dec. 3.
>
> **(2) Weakness**: “To me, the study reveals how closely LLMs can replicate the annotations of legal annotators on LLM-generated material. The truly interesting question would be whether LLMs can identify genuine non-compliance among real developers and deployers using authentic, non-pre-selected data. This question cannot be answered with the current study design and I find the result of the paper very limited in novelty and impact.”
>
> **Response**: Unfortunately, as described in Sec. 2, real samples of technical documentations are likely subject to confidentiality and well as legal privilege. This may explain why none have been shared publicly (e.g., on the web) thus far. That said, as already discussed in our Future Work (Sec. 7), we are interested in helping owners of this type of data anonymize it so that it can be used in a benchmark like this. To make this clearer for future readers, we have improved the language of Sec. 7.
>
> **(3) Weakness**: “The paper appears to have been developed with legal input, yet it lacks a clear legal interpretation framework or taxonomy of expected non-compliance types. This omission further weakens external validity, as the instances of non-compliance planted in the LLM-generated excerpt originate from the LLM process itself. Including even a brief qualitative analysis of the most frequent or illustrative non-compliance patterns would substantially strengthen the work's legal validity.”
>
> **Response**: To address this point, we are currently working on an analysis of the patterns observed in the legal annotation justifications — for example, the recurring compliance violation patterns. For this, we have relied on the qualitative free text justification of the Likert scores, described in Sec. 3.  We aim to add this analysis to the paper by the end of the discussion period on Dec. 3.
>
> **(4) Weakness**: “I am not fully convinced by the novelty claim that this is the first benchmark of LLMs for AI regulation compliance. For example, Guldimann et al. (2024) already present an entire benchmarking suite for AI Act compliance of LLMs. The authors should more clearly position their work relative to such prior efforts.”
>
> **Response**:  There is an important distinction between works like Guldimann et al. (2024) and this one. Guldimann et al. (2024) create a benchmark that evaluates whether LLM-based AI systems can comply with the AIA. Differently, this work here creates a benchmark (we believe the first) that evaluates whether LLMs can perform compliance assessments. To avoid further confusion, we have added a new subsection (5.4) to our Background and Related Work section that further clarifies this distinction.
>
> **(5) Weakness**: “The introduction and motivation would benefit from concrete examples of the types of violations assessed.”
>
> **Response**:  To address this point, as mentioned, we're currently working on an analysis of the patterns observed in the legal annotation justifications — for example, the recurring compliance violation patterns. We aim to add this analysis to the paper by the end of the discussion period on December 3. In the meantime, we have also added Appx. J, which contains a full example of a sample and the compliance scores assigned to it by our expert annotators.
>
> **(6) Weakness**: “The evaluation design using the median human annotator as a reference point is not ideal. In related literature, stronger baselines are common. I recommend that the authors consider applying the Alternative Annotator Test proposed by Calderon et al. (2025).”
>
> **Response**:  At your suggestion, we've added the results of the test in Calderon et al. (2025). Please see Table 8 in Appx. B.
>
> **(7) Weakness**: “Minor: typo line 12: Systems Systems; and 1153 )) instead of )”
>
> **Response**: Fixed.

---

> > ### Author Response · Authors · 2025-11-22
> > **Initial response to Reviewer AXmy (Part II)**
> >
> > Thanks again for your thoughtful feedback. As promised, here is Part II, with responsers to the Questions you've raised:
> >
> > **(1) Question**: “Why did you use gpt-4.1-mini for the technical documentation generation?”
> >
> > **Response**: This particular model was chosen because of its affordable price point as well as the fact that, during our initial testing phase, it was deemed to produce plausible results. We also experimented with o3-mini, 4o-mini, and gpt-5 later but found that, for this use case and for our pipeline, these models were not as adept at following our instructions and producing plausible outputs. We have added the fact that we considered these other models to the paper, and explained why we ultimately chose gpt-4.1-mini, in Sec. 2.  Please see the updated PDF.
> >
> > **(2) Question**: “Can compliance with the selected AIA articles truly be assessed from technical documentation alone?”
> >
> > **Response**: In some cases, we believe the answer is yes. As described in Appendix D, multiple compliance experts that we consulted described technical documentation as the main deliverable to the compliance assessor in a review. That said, as we point out in Sec. 7, the assessment is sometimes the result of a multi-turn conversation and may draw on a wider body of evidence; this is something we are excited to pursue as Future Work.
> >
> > **(3) Question**: “Which articles/violation caused the highest annotator disagreement?”
> >
> > **Response**: Quantitatively, the highest average disagreement occurred in Article 10 (Data and Data Governance) and Article 15 (Accuracy, Robustness, and Cybersecurity), with mean standard deviations of 0.638 and 0.612 respectively. The most severe disagreement revolved around the intended use of Credit Scoring (Use  8) under) Article 9, which received scores of (1, 5, 5), resulting in the dataset's highest standard deviation of 1.89. We have added these details to Sec. 3.1.
> >
> > **(4) Question**: “Are differences between models statistically significant?”
> >
> > **Response**: In short, yes. We determined statistical significance by performing pairwise bootstrap resampling (N=2,000) on the models' Cohen's Kappa scores, The analysis shows performance tiers, with a top group (Gemini 2.5 Pro, GPT-5, Grok 4) demonstrating a statistically significant performance advantage over nearly all mid- and low-tier models. We have added these new insights to the paper, as Table 7 in Appendix B.
> >
> > **(5) Question**: “Why were no open-weight models included in the evaluation?”
> >
> > **Response**: To clarify, we did indeed include some open-weight weight models in the evaluation (see Appendix B, Table 6). Specifically, we evaluated open-source legal LLMs Saul-7B-Instruct (Colombo et al., 2024b) and Saul-54B-Instruct (Colombo et al., 2024a).
> >
> > **(6) Question**: “Since legal annotators could use external sources, did any rely on LLMs for assistance? Some justifications in the repo appear LLM-like to me.”
> >
> > **Response**: Outside of the instructions in Appendix F, annotators were specifically instructed not to use LLMs to write these justifications (or, more generally, to produce their annotations). We have added this as a footnote in Appendix F. As described in Appendix F, to provide some degree of structure, we asked annotators to begin their justifications with the same boilerplate introduction; it is possible this was the cause of the phenomenon you describe.
> >
> > **(7) Question**: “Given the technical nature of the documents, is this annotation task really best suited to legal experts?”
> >
> > **Response**:  We believe it is. This is because, ultimately, it is the compliance assessors, including legal experts who must analyze these technical documents and, mapping them onto the text of the law, render a compliance analysis . While some of these compliance assessors may include non-lawyers, the skillset of a legal expert (mapping facts onto law) is an optimal proxy.
> >
> > **(8) Question**: “Have you considered that, in real use, developers might try to game such LLM-based compliance systems?”
> >
> >
> > **Response**:  Although we intended to flag this risk in Sec. 6.4, we have updated the language in this section to make this more obvious. Please see Sec. 6.4 in the updated PDF. We have also added this to future work in Sec. 7.

---

### Official Review · Reviewer_D1j8 · 2025-11-05

**Soundness:** 4
**Presentation:** 2
**Contribution:** 4
**Rating:** 10
**Confidence:** 5

**Summary:**

The goal of the paper is to address the issue of there being no standardized method for quantitatively evaluating and comparing the performance of LLMs for AIR compliance assessments through their open dataset AIReg-Bench.

**Strengths:**

The strengths of the paper is that is addresses an issue of high interest and significance, the ideas presented are original, the paper is very clear and straightforward to read and follow, and sufficient details are provided for claims that the authors make. Also, the steps listed for reproducibility is provided correctly. Overall, it is an excellent paper in terms of scientific contributions, impact, importance and necessity.

**Weaknesses:**

One of the weaknesses of the paper is the presentation, such as how it is structured. The better way in which the paper should have been structured is first the problems are detailed, followed by a brief introduction to the solution and then thorough details about the solution. After the details about the solution are provided, the paper should then continue with explaining how the solution addresses the problem and provide evidence/support accordingly. The worry I have is that the presentation of the paper could diminish the significance and contributions of the paper.

Another weakness is the types of people that were involved in the study, When the authors say legal experts, they mention how those legal experts are a team of law school students, law graduates, and qualified lawyers. The problem is that law school students are not legal experts. There is still a lot that they need to do before they become experts in law. In some extent the same applies for law graduates, but law graduates is more acceptable than law students. I understand the benefit of having a diverse set of legal individuals, but this could hinder the quality of the paper.

**Questions:**

What is the legal practice of the lawyers? There is a big difference in knowledge, approach and perspectives on issues between a criminal lawyer, family lawyer, and an immigration lawyer.

If the paper is not accepted to the conference, I would suggest performing the study again with just qualified lawyers. It would make the paper much more valuable.

Did an ethics board review and approve the study? If so, this needs to be added in the ethics statement. Also, the ethics statement could be worded better.

---

> ### Author Response · Authors · 2025-11-22
> **Initial response to Reviewer D1j8**
>
> We thank you for your thoughtful and constructive feedback. Your 10/10 score, the highest that can be awarded, is very encouraging.
>
> Below, we respond to each of the Weakness and the Questions raised, which we have copied below for your convenience. As you will hear, we have updated the paper in accordance with your feedback, often adding new material, and encourage you to review the updated PDF attached to this submission.
>
> **(1) Weakness**: “One of the weaknesses of the paper is the presentation, such as how it is structured. The better way in which the paper should have been structured is first the problems are detailed, followed by a brief introduction to the solution and then thorough details about the solution. After the details about the solution are provided, the paper should then continue with explaining how the solution addresses the problem and provide evidence/support accordingly.”
>
> **Response**: Thank you for your suggestions. While we believe that the structure of our work already, in large part, aligns with your suggestions. However, we have added some sign-posting to make this clearer. For example, we have changed the title of our Sec. 1, which is meant to introduce the problem statement, from “Introduction” to “Introduction and Problem Statement” (Sec. 1). We have also changed the titles of Sec. 2 and 3 to reflect the fact that our proposed solutions can be found there. Finally, we have changed the title of Sec. 4 to reflect the fact that the Experiments described therein seek to prove that our solutions indeed address the problem.
>
> **(2) Weakness**: “Another weakness is the types of people that were involved in the study, When the authors say legal experts, they mention how those legal experts are a team of law school students, law graduates, and qualified lawyers. The problem is that law school students are not legal experts. There is still a lot that they need to do before they become experts in law. In some extent the same applies for law graduates, but law graduates is more acceptable than law students. I understand the benefit of having a diverse set of legal individuals, but this could hinder the quality of the paper.”
>
> **Response**: To clarify, our team of legal expert annotators consisted of two (2) qualified lawyers, three (3) law school graduates, and just one (1) law school student. We have added these details about the group to the paper, as Appendix I.
>
> Regarding the sole law school student on our team, as highlighted in Sec. 3.1, there is precedent for using law school students as legal expert annotators, including in papers accepted to top tier conferences. This includes Ostling et al., 2024 (NeurIPS 2024), Shen et al., 2022 (NeurIPS 2022), Hendrycks et al., 2021b (NeurIPS 2021), Wang et al., 2025b (ACL 2025); Zheng et al., 2025 (CSLAW 2025), Wang et al., 2023 (EMNLP 2023), and Zhong et al., 2019 (ICAIL 2019). Because a key skill taught and acquired in law school is the ability to map an arbitrary legal text onto a set of facts, and because that is the heart of the annotator analysis here, we would politely argue that this approach holds water.
>
> **(1) Question**: “What is the legal practice of the lawyers? There is a big difference in knowledge, approach and perspectives on issues between a criminal lawyer, family lawyer, and an immigration lawyer. If the paper is not accepted to the conference, I would suggest performing the study again with just qualified lawyers. It would make the paper much more valuable.”
>
> **Response**: We have added our annotator specializations, where applicable, as Appendix I. As you will see, all of them listed AI regulation — or regulation more broadly — as one of their specializations. We have also added this fact to Sec. 3.1 of the paper.
>
> **(2) Question**: “Did an ethics board review and approve the study? If so, this needs to be added in the ethics statement. Also, the ethics statement could be worded better.”
>
> **Response**: We have augmented and tried to improve the wording of the Ethics Statement; please see Sec. 9 in the updated PDF. Because our human legal expert annotators are all co-authors on the paper as well, no ethics board review was deemed necessary in this instance and, as such, none has been attached to the Ethics Statement.

---

### Author Response · Authors · 2025-11-22
**Summary of improvements made in response to initial feedback**

We thank the reviewers for their thoughtful and constructive feedback. We will respond directly to each reviewer about the specific Weaknesses and Questions they raised in their reviews.

As everyone can see, in response to their feedback, we have made some significant improvements to the paper and plan to make additional significant improvements before the conclusion of the discussion phase on December 3.

The improvements to the paper we have already made, and which are shown in blue in the updated PDF attached to this submission, include:

1. **Sec. 2:** An explanation of why gpt-4.1-mini was chosen over other models for our generation pipeline.
2. **Sec. 2:** Further explanation of why we chose to scope the dataset to a subset of the EU AI Act's requirements for high-risk AI models.
3. **Table 1:** A new table providing summary statistics for our legal expert annotator's plausibility scores.
4. **Sec. 3:** A deeper analysis and more statistics on inter-rater agreement.
5. **Sec. 5.4:** An entirely new subsection discussing prior works involving non-LLM algorithms that assess AI regulation compliance, helping differentiate this work.
6. **Sec. 6.4:** Further discussion of the risks of AI developers “gaming” LLM compliance assessors.
7. **Sec. 7:** Additional discussion of proposed future work, including extending the dataset to real-world documentation and to multi-turn interactions.
8. **Sec. 9:** An augmented Ethics Statement.
9. **Table 7 (Appendix B):** A new table showing pairwise Cohen’s Kappa scores comparing model performance.
10. **Table 8 (Appendix B):** A new table displaying the results of the Alternative Annotator Test (Calderon et al., 2025) on our legal expert annotations.
11. **Appendix I:** A new appendix detailing our team of legal annotators’ expertise and specializations.
12. **Appendix J:** A new appendix containing a full sample technical documentation and annotations that show how it was annotated by both legal experts and an LLM annotator.

The additional improvements we plan to make before the conclusion of the discussion phase on December 3 are:

1. **AI Developer Plausibility Annotations:** We have recruited industry AI developers to provide additional external validation of all samples in our dataset via plausibility scores and justifications. This will add a new perspective that augments the existing legal expert plausibility scores. We aim to complete and integrate this new set of annotations into the paper by December 3.
2. **Dataset Augmented with LLM-Annotated Samples from Additional Models:** By December 3, we plan to add a new component to our dataset that includes samples generated by additional models (beyond gpt-4.1-mini) and annotated by fine-tuned LLMs trained on annotator text justifications. Although this synthetic data will not be annotated by human legal experts, and therefore very different than what exists today, it will materially expand the dataset's size and diversity and, we believe, will serve as a foundation for future expert annotation and experimentation.
3.  **Deep Analysis of Compliance Violation Patterns:** By December 3, we aim to add an in-depth analysis of patterns in legal expert compliance scores—for example, recurring compliance violation patterns noted in annotator justifications.
4. **Updated Anonymous GitHub URL:** By December 3, we aim to update and re-anonymize our GitHub repository with all the new code changes that drove the additions described above. Until then, the anonymous GitHub URL in the paper contains the code as it existed before these latest changes.

---

> ### Author Response · Authors · 2025-12-02
> **Final improvements made in response to initial feedback**
>
> In the end, in time for the conclusion of this discussion period, we were able to complete the following items from our list of planned improvements: item 3 (Deep Analysis of Compliance Violation Patterns) and item 4 (Updated Anonymous GitHub URL). The Deep Analysis of Compliance Violation Patterns has been added as Appendix K (also referenced in Sec. 3.1) and the updated anonymized repo been updated at the URL and also as a .zip attached to this submission. Please see the updated PDF reflecting the new changes and thank you once again for your time.

---

### Meta-Review · Area_Chair_deVk · 2026-01-06

**Summary:**

Reviewer pointed out concerns regarding paper structure and presentation style, novelty of the work, methodological rigour, particularly pertaining inter-annotator agreement, disagreements patterns and statistical significance of model differences. More fundamentally, concerns pertaining to the use of synthetic technical documentation and how LLM-generated violations may reflect LLM artifacts rather than real-world non-compliance have been informative to my decision.

**Reviewer Concerns:**

Concerns pertaining paper structure, novelty as well as typos were satisfactorily addressed. Concerns relating to inter-rater agreement and statistical significance were partially addressed. Unfortunately, although the authors have provided justification, lack of real data undermines the contribution of the paper. Reliance on synthetic data as well as lack of external validity remain a major limitation.

**Reviewer Scores:**

This paper received one extremely enthusiastic review with a Rating of 10, two mostly critical reviews (Rating 2), and a modest review (Rating 4) with an average Rating of 4.50 and average confidence of 4.25.

---

### Decision · Program_Chairs · 2026-01-26

Reject